# An integrative molecular map of pediatric B-cell precursor acute lymphoblastic leukemia

Olga Krali [1,2], Anna Pia Enblad[1,2,3], Julia Sulyaeva[1,2], Dea Gogishvili[1,2], Anders Lundmark[1,2], Arja Harila[3], Claes Andersson[1], Tom Erkers[4,5], Merja Heinäniemi [6], Gudmar Lönnerholm[3] & Jessica Nordlund [1,2] ✉

## Abstract

**Background** The molecular landscape of pediatric B-cell precursor acute lymphoblastic leukemia (BCP-ALL) has been extensively characterized through single-modality studies. However, the interplay between molecular modalities and their collective influence on treatment response and outcomes remains poorly understood.

**Methods** We integrated genomic, epigenomic, transcriptomic, and ex vivo drug response data from 1231 patients diagnosed with BCP-ALL. Using Multi-Omics Factor Analysis, we identified signatures explaining key aspects of the integrative molecular landscape, referred to as cross-modal elements (CMEs). The CME-derived signatures were introduced into pathway and intermodal network analyses, while their impact on patient outcomes was assessed through survival modeling.

**Results** Pathway and network analyses annotate the resulting integrative CMEs, linking them to key biological processes, including disease development, cellular regulatory processes, metabolic pathways, and drug response. By leveraging correlations between DNA methylation and ex vivo response to doxorubicin, we stratify patients with hyperdiploidy into subgroups that differ in relapse-free survival. These signatures are independent of clinical variables. Survival models incorporating CME-selected ex-vivo drug responses combined with clinical data improve risk prediction compared to clinical models alone (FDR < 0.05), demonstrating the potential of integrative multiomics in refining risk stratification.

**Conclusions** Our study highlights the importance of multimodal data integration in BCP-ALL to provide biological insights with potential relevance for precision medicine.

## Plain language summary

Pediatric BCP-ALL is the most common childhood cancer. While most children are cured today, some still relapse, die from the disease, or experience severe treatment-related side effects. To better understand why outcomes differ, we analyzed data from 1,231 children with BCP-ALL, integrating multiple layers of information from their leukemia cells. This approach revealed biological patterns related to leukemia subtype, cell growth, and treatment response. We also found that including information on how leukemia cells respond to treatment improved relapse prediction compared with using clinical data alone. Together, our findings show that combining different types of data can reveal subtle differences between patients and help us better understand the complex biology of childhood leukemia.

Over the past decades, the molecular phenotypes underlying pediatric B-cell precursor acute lymphoblastic leukemia (BCP-ALL) have been extensively studied[1]. Strong associations between genetic subtypes and corresponding molecular phenotypes have been well established[2–4]. These associations extend beyond recurrent somatic single-nucleotide variants (SNVs)[2] to include quantitative signatures derived from DNA methylation (DNAm)[5–7], gene expression (GEX)[8–10], protein expression[11–13], and metabolomics data[14]. Collectively, these studies have significantly expanded our understanding of ALL pathogenesis, enabling improved risk stratification and therapy monitoring, which have contributed to the remarkable cure rates achieved

in recent years[1]. Despite cure rates exceeding 90% for most BCP-ALL patients[15–17], significant challenges remain. Relapsed patients have poor outcomes, with survival rates of ~50%[17,18], while others suffer severe side effects and morbidities resulting from overtreatment[16].

Most studies focus on single molecular entities or modality comparisons, despite the fact that molecular drivers of BCP-ALL, like those in other cancers[19,20], are likely to operate across multiple interconnected molecular levels[21]. Several studies have successfully integrated more than one data modality[11,22–27]. Meanwhile, ex vivo drug response (EVDR) profiling is emerging as a promising precision medicine tool, providing direct

[1]Department of Medical Sciences, Uppsala University, Uppsala, Sweden. [2]SciLifeLab, Uppsala University, Uppsala, Sweden. [3]Department of Women's and Children's Health, Uppsala University, Uppsala, Sweden. [4]Department of Oncology-Pathology, Karolinska Institutet, Stockholm, Sweden. [5]SciLifeLab, Stockholm, Sweden. [6]Institute of Biomedicine, School of Medicine, University of Eastern Finland, Kuopio, Finland. ✉e-mail: jessica.nordlund@medsci.uu.se

assessment of how patient-derived cells respond to established and experimental therapies[28–31]. Bridging the gap between single-modality and integrative, multimodal approaches is essential for deepening our understanding of ALL pathogenesis and accelerating precision therapies.

Several computational methods for inter-modal integration have been developed to better understand complex disease patterns in cancer[22,25–27,32,33]. Among these, Multi-Omics Factor Analysis (MOFA) is an unsupervised framework that integrates diverse data modalities to identify functional multiomics patterns[22]. In chronic lymphocytic leukemia (CLL), MOFA revealed latent factors linked with multimodal heterogeneity, identifying previously overlooked pathways, such as oxidative stress response, as key contributors to cellular processes, disease development and progression, and clinical outcomes[22]. In Myelodysplastic syndrome (MDS), MOFA identified potential prognostic markers and protective mechanisms associated with patient outcome[26]. In ALL, multiomics integration has been applied to cell-lines[11] and data from T-ALL patients[27], however, such integration has yet to be been applied to data from primary BCP-ALL patient samples.

This study provides an integrative molecular map of pediatric BCP-ALL, combining SNVs, DNAm, GEX, and EVDR data from 1231 patients. By correlating these data with clinical outcomes, we identify cross-modal elements (CMEs) that capture shared heterogeneity, which we interpret in the context of BCP-ALL pathophysiology. Intermodal network analysis detects DNAm signatures with potential for patient stratification, revealing a subgroup of patients with inferior outcomes. Adding CME information to risk models demonstrates its potential for linking functional and molecular signatures to improve risk stratification.

## Methods
### Patients
Treatment-naive peripheral blood or bone marrow aspirates were obtained from 1231 pediatric patients with BCP-ALL at the time of diagnosis with blast counts >80%. The children were diagnosed between 1992–2013 and treated according to the Nordic Society of Pediatric Hematology and Oncology (NOPHO) NOPHO ALL-92 ($n = 303$), ALL-2000 ($n = 504$), ALL-2008 ($n = 367$), EsPh-ALL ($n = 17$), or Interfant ($n = 40$) treatment protocols[16,34–36]. The median age of the patients at diagnosis was 4.5 years (interquartile range, IQR, 2.9–8.4), and the median follow-up time was 17 years (IQR, 13–21). Cytological analysis, immunophenotyping, and cytogenetics of pretreatment leukemic cells enabled the establishment of the molecular diagnosis of BCP-ALL. Molecular subtypes were revised to the current international consensus classification (ICC) as previously described[37]. Guardians and/or patients provided written informed consent. The study complied with the Declaration of Helsinki and was approved by the NOPHO Scientific Committee (Study #56) and the Regional Ethics Review Authority (reference numbers: DNR 2007/023, 2010/416, 2013/237, 2014/482) in Uppsala, Sweden.

### DNA methylation (DNAm)
The methylation status was interrogated using the Infinium HumMeth450K BeadChip assay (450k array, $n = 1012$ patients) or the MethylationEPIC v2.0 BeadChip assay (EPIC array, $n = 55$ patients). The data were processed as previously described[5,37,38]. Probes located on the X and Y chromosomes and probes known to be affected by underlying genetic variation were filtered out[5]. Only probes overlapping the two array types (450k array and EPIC) were retained, resulting in 372,264 CpG sites across 1067 patients for downstream analysis. For copy number analysis, the methylumi and conumee[39] 2.0 R packages were used.

### RNA sequencing (GEX)
RNA sequencing (RNA-seq) GEX data were retrieved from GSE227832 and processed as previously described[37]. Only protein-coding genes were retained. Additionally, genes located in X and Y chromosomes, as well as ribosomal, mitochondrial, and scaffold genes were removed. The final GEX dataset contained protein-coding 18,928 genes in 295 patients for downstream analysis. The *limma*[40] R package was used to perform differential gene expression analysis.

### Single nucleotide variants (SNVs)
Mutational data (somatic SNVs) from 128 patients were retrieved from previous studies[37,41–44]. In short, the somatic SNV data were generated either from an 872-cancer gene Haloplex panel[41,42] ($n = 125$) or/and from whole genome sequencing (WGS, $n = 16$)[41,43,44]. For the patients with RNA-seq data available ($n = 295$) the variant alleles *PAX5* p.Pro80Arg, *IKZF1* p.Asn159Tyr, and *ZEB2* p.His1038Arg were specifically interrogated as previously described[37]. In total, SNV data across 529 genes and 128 patients were available for downstream analyses.

### Ex vivo drug response (EVDR)
Ex vivo drug response to a panel of ten treatment compounds was assessed using the Fluorometric Microculture Cytotoxicity Assay (FMCA) as previously described[45]. The data represent the fraction of surviving leukemic cells after 72 hours of incubation with each drug at empirically selected concentrations (Supplementary Data S1). These concentrations were chosen to yield survival index (SI) values that capture inter-sample variability[38]. The SI was calculated as the mean fluorescence signal from wells containing leukemic cells with intact plasma membranes after drug exposure, divided by the mean fluorescence signal from wells containing untreated leukemic cells (control), after subtracting the background signal from blank wells (medium only). This ratio was multiplied by 100 to obtain SI%[45].

The drugs belong to the glucocorticoid (dexamethasone and prednisolone), topoisomerase II inhibitor (amsacrine, doxorubicin, mitoxantrone, and etoposide), antimetabolite (cytarabine and thioguanine), enzyme (L-asparaginase), and vinca alkaloid (vincristine) classes. In total, EVDR data for 857 patients across 10 drugs were available for downstream analyses.

### Multi-omics factor analysis
The data containing all four data modalities (DNAm, GEX, SNV, and EVDR) were initially split into training ($n = 923$) and test ($n = 308$) datasets in a stratified manner to retain the data modality proportions of the entire cohort. Patients treated on different NOPHO treatment protocols were evenly distributed across the two datasets. MOFA was employed using the mofa2 R library[22] on the training dataset. We renamed the MOFA factors as CMEs. To run MOFA, we selected a subset of CpG sites ($n = 5000$) and genes ($n = 10,000$) with the highest variance in the training dataset, and included all of the available SNV ($n = 529$) and EVDR ($n = 10$) features. This variance-based feature selection follows MOFA recommendations to reduce dimensionality and exclude low-variance features. Because RNA-seq data were available for only 24% of patients compared to 87% for DNA methylation, we included a larger number of genes to balance the relative contribution of the transcriptomic modality and prevent overrepresentation of methylation data. The training dataset was used to create a MOFA object and run the analysis, without imputation. The train options were kept as default apart from the convergence mode (medium instead of slow), the number of maximum iterations (700 instead of 1000) and the variance explained threshold (2% instead of none). A seed parameter of 42 was applied when running the analysis. MOFA generates CME values at a patient level and feature weights at a feature level for all CMEs. We extracted the features weights scaled from −1 to 1 at a CME and modality level, where −1 represented the highest negative impact, 1 the highest positive impact, and 0 no impact on each CME. For the GEX and DNAm modalities specifically, we selected the top 1000 genes and CpG sites (based on the highest absolute weights) per CME for downstream analyses. MOFA was used to impute missing values in the training set. A separate MOFA model was built for the test set using the same parameters as the training model, applied to impute missing values independently.

### Annotation of CpG sites

The GREAT 4.0.4 annotation tool[46] was used to annotate CpG sites to genes in proximity based on the distance from a transcription start site (TSS) extended by 5 kb upstream and 1 kb downstream, and up to 10 kb extension in both directions, including regulatory domains curated from literature. For each CME and weight (positive, negative), a bed file was generated containing the start and stop coordinates, and the chromosomal location of each CpG site. A bed file was prepared for the background CpG sites ($n = 5000$ CpG sites).

### Pathway analysis

Over-representation analysis (ORA) was performed on the top-ranked CpG sites and genes derived from the MOFA model trained on the training dataset, with absolute weight >0.6 for each CME using the enrichr()/enrich() modules from the gseapy Python library[47]. The positively and negatively weighted features were analyzed separately. The background was set as the top 5000 annotated CpG sites (1735 gene annotations) and 10,000 ENSEMBL genes (9998 unique gene names), which were used as the input features for MOFA. Each CME/weight gene list was tested on gene sets from gene ontology (GO) biological processes (BP), the molecular signatures database (MSigDB) Hallmark, and seven custom gene lists. The custom lists comprised subtype classifiers, as ALLIUM DNAm (272 annotated CpGs) and GEX (356 genes)[37], ALLSorts[48] (1003 genes) and ALLCatchR[49] (2,802 genes), cell cycle genes from Seurat[50] (97 genes), and cell state genes from bulk (145 genes) and single cell RNA-seq (574 genes) ALL samples[51].

Pathway networks and clusters were generated using the enrichment map function of gseapy[47]. Pathways with ≤1 overlapping gene, as well as CMEs with ≤2 pathways in present, were excluded from the analysis. To annotate the resulting clusters, we performed text mining by tokenizing the pathways into individual words using CountVectorizer from scikit-learn[52] using the 20 most frequent words per cluster.

### Network analysis

The igraph R library[53] was used to build inter-modal networks. For each CME, the most impactful features (absolute weight >0.6) were retrieved for both training and test datasets. Networks were generated from the training data, and the reproducibility of central dependencies was assessed in the test data. The features with positive and negative weights were analyzed separately. Pairwise correlation matrices (DNAm-GEX, DNAm-EVDR, GEX-EVDR) were generated using the non-imputed data, ensuring the analysis reflects real correlations without bias due to imputation. For each CME, an absolute correlation coefficient cut-off (0.2–0.6) was applied to remove weak correlations among features. The correlation matrices were combined to create an adjacency matrix, which was used as input to create the network graphs. Self-loops, as well as unconnected nodes, were removed from the networks. A random seed 1 was employed for each plot to control the stochasticity of the graphical representations. Finally, the resulting networks were used to extract communities (cluster_edge_betweenness) and hub features. Each network consisted of one or multiple clusters (communities) of features. Within these networks, the features with > 3 connections (correlations) were assigned as central points (hubs).

### Survival models

Gradient boosted Cox proportional hazards loss survival models[54] were built using the MOFA-imputed training set in three configurations. A CME feature-based (CMEf, top features with absolute weight >0.6) model, a baseline model trained on clinical risk groups, and a combined model incorporating both CMEf and baseline features. In addition, single-omics (DNAm, GEX, and EVDR) and the CMEf-all features model were constructed. These models were also evaluated combined with the baseline features. This resulted in a total of 83 different models: one clinical baseline model, two CMEf-all models (with and without clinical data), and 20 models each based on top-ranked features from the multiomic (CMEf), DNAm, GEX, and EVDR data, each evaluated alone and in combination with clinical risk grouping.

The survival models link the covariates to the time of an event, which, in our case, corresponds to event free survival (EFS). They constitute a type of regression model, which generates risk scores in relation to an event (event vs. no event). In our cohort, 227 and 81 events occurred in the training and test sets, respectively (relapse, death in complete remission, induction failure, secondary malignant neoplasm, and resistant disease as defined in the protocol in question). EFS is calculated from the time at diagnosis to the occurrence of an event or, for patients who did not experience an event, to the date of the last follow-up.

Each survival model comprised 100 regression trees. Repeated stratified k-fold cross-validation (CV) was performed[52] using $3 \times 5$ repeated splits. A stratified setup retains the same event distribution for the inner training and validations CV sets, as in the outer training dataset ($n = 923$). A CV setup mitigates overfitting, as each model is trained on the inner training sets and is validated on the left-out (CV) data. The CV scores were extracted to perform statistical analysis using BH-adjusted Wilcoxon signed-rank test to assess differences in model performance between the baseline and the molecular models. The hyperparameter space included the model's learning rate (0.01), the number of minimum samples to split a tree node (5), the maximal depth of the tree (5), and the maximal number of features for the best node tree split (square root of #features). As the survival models predict risk scores, Harrell's concordance index (c-index) was used to evaluate their performance. The c-index takes into account the magnitude of the assigned risk in relation to the time that an event has occurred. For instance, a model with a high c-index assigns higher scores to patients who experience an event at a shorter time span. In addition, the models were further validated using the MOFA-imputed test set.

### Statistics and reproducibility

All statistical analyses were performed using Python (version 3.8.5) or R (version 4.2.3).

Where applicable, Pearson's correlation coefficients were used to assess relationships between two variables, with corresponding $p$-values adjusted for multiple testing using the Benjamini-Hochberg procedure (FDR < 0.05).

Enrichment analyses (ORA) relied on hypergeometric tests to obtain significantly enriched pathways (FDR < 0.01).

Pairwise group comparisons were made using two-sided Mann-Whitney U tests. For comparisons involving more than two groups, Dunn's test was applied. Multiple testing correction was performed using the BH method (FDR < 0.05).

Kaplan–Meier survival curves were compared using log-rank test ($p$-value < 0.05). Multivariable Cox proportional hazards models were fitted to estimate hazard ratios, and statistical significance was assessed using the Wald test ($p$-value < 0.05). Fisher's exact test was used to assess differences in proportions of good- or poor-risk patients between the low- and high-doxorubicin response clusters ($p$-value < 0.05).

Two-sided Wilcoxon signed-rank tests were used to evaluate cross validation performance between the baseline clinical model and the CME-based models, followed by BH correction for multiple testing (FDR < 0.05).

Sample sizes varied depending on data availability across molecular modalities and are reported in the corresponding Results sections, figures, or figure legends. No experimental or technical replicates were included in this study. All analyses were conducted using predefined statistical thresholds as described above.

## Results

### Cohort demographics and data overview

Pre-treatment peripheral blood samples or bone marrow aspirates were retrieved from 1231 children diagnosed with BCP-ALL between 1992 and 2013, who were enrolled in the consecutive NOPHO ALL-92 ($n = 303$), ALL-2000 ($n = 504$), ALL-2008 ($n = 367$), EsPh-ALL ($n = 17$), or Interfant treatment protocols ($n = 40$)[16,34–36].

The omics data were obtained from previous studies: Genome-wide DNAm profiling data were obtained from 1,067 patients using Illumina 450k ($n = 1012$)[37] or EPIC ($n = 55$) arrays[38]. RNA-seq data were available

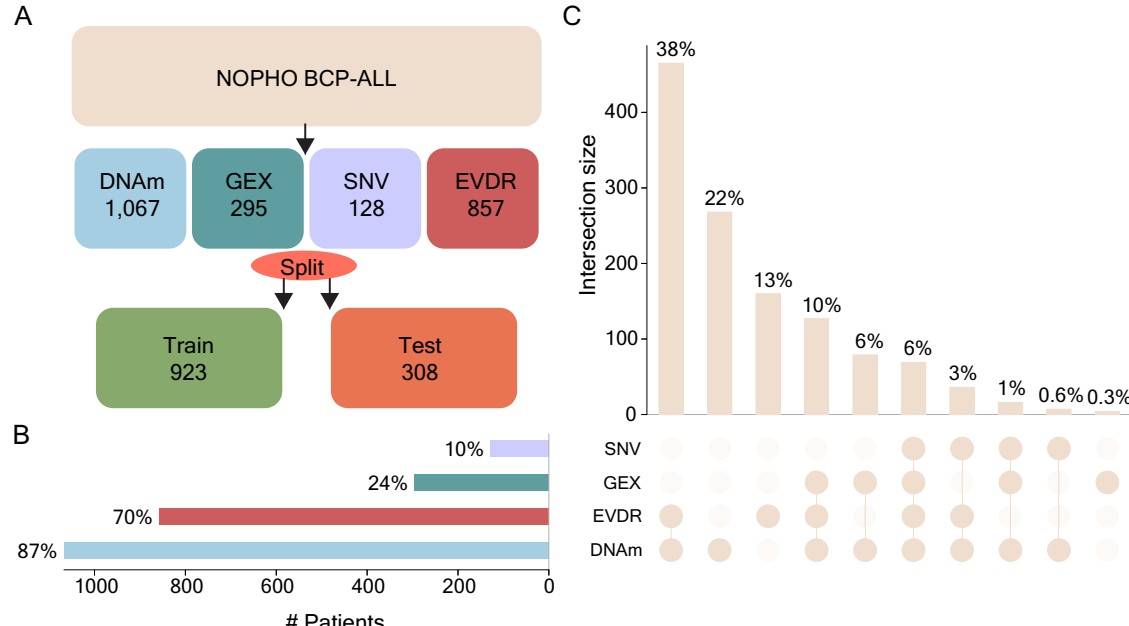

**Fig. 1 | Pediatric BCP-ALL samples and data modality overview. A** DNA methylation (DNAm), gene expression (GEX), somatic single-nucleotide variants (SNVs), and ex vivo drug response data (EVDR) from 1231 pediatric BCP-ALL patients. The patients were split into training (75%) and test sets (25%). **B** Percentage-wise sample distribution across each modality. **C** Upset plot showing the matched samples between two, three, all, or none of the data modalities.

from 295 patients[37]. Somatic SNVs were obtained using a 872-cancer gene Haloplex panel (*n* = 125 patients)[41,42], WGS (*n* = 16 patients)[41,43,44], or by targeted analysis for single-nucleotide variants *PAX5* p.Pro80Arg, *IKZF1* p.Asn159Tyr, and *ZEB2* p.His1038Arg[37] (*n* = 295 patients). EVDR data for 10 drugs were generated with the FMCA[45] for 857 patients (Fig. 1A).

The dataset was split into training (*n* = 923, 75%) and test (*n* = 308, 25%) datasets stratified to balance the representation of DNAm, GEX, SNV, and EVDR data (Fig. 1A, B, Fig. S1). The percentage of matched samples varied, with 65% of samples appearing in at least two modalities, 20% in at least three modalities, and 6% in all four modalities (Fig. 1C).

**Cross-modal elements are associated with clinical features**

We applied MOFA[22] on the training dataset (*n* = 923 patients) using four data modalities: DNAm, GEX, SNV, and EVDR (Fig. S1, Supplementary Data S2). For DNAm and GEX, we retained the 5000 and 10,000 most variable features, respectively, while all dimensions were included for SNV (*n* = 529) and EVDR (*n* = 10) data (Fig. 2A). This analysis generated a matrix (Z) of CME values per patient and a feature weight matrix (W[1],…, W[M]) for each CME and data modality. MOFA identified 10 CMEs using a total variance explained cut-off of 2%, explaining 55% of variance in DNAm, 48% in GEX, 24% in EVDR, and 0% in SNV data (Fig. 2B). CMEs were categorized as either modality-specific (e.g., CME 3–4, 8, and 10) or inter-modal (e.g., CME 1–2, 5–7, and 9, Fig. 2C), based on a variance explained cut-off of >1%. For modality-specific CMEs, CME 3 explained 17% of the variance in the DNAm data, while CMEs 4 (14%), 8 (8%), and 10 (3%) explained the most variance in the GEX data. Inter-modal CMEs (1–2 and 5–7) captured shared variability across DNAm, GEX, and EVDR data, while CME 9 captured variability shared between DNAm (6%) and GEX data (2%). None of the CMEs captured variability in the SNV data, likely due to a combination of the small number of recurrent SNVs in ALL[55] and the sparsity of our dataset[42]. Consequently, SNVs were excluded from downstream analyses.

To evaluate model robustness and CME orthogonality, pairwise correlations between CMEs were calculated, revealing no significant codependence between CMEs (Fig. 2D, absolute rho <0.25). Next, we assessed correlations between CMEs and clinical covariates, including sex, age at diagnosis, outcome, treatment protocol, risk group, and molecular subtype (Fig. 2E). No strong association was observed for sex

and outcomes across CMEs (absolute rho < 0.15, Fig. 2E, Supplementary Data S3). CMEs 4 and 9 were correlated with age at diagnosis (rho = −0.34, FDR < 0.001 and rho = 0.40, FDR < 0.001, Fig. 2E, Supplementary Data S3). Although it is well established that age at diagnosis is correlated with molecular subtype[56], no significant correlation was observed between CME 4 and any of the molecular subtypes or risk groups (Fig. 2E). However, CME 9 was positively correlated with the PAX5alt subtype (rho = 0.46, FDR < 0.001, Fig. 2E). Additionally, patients treated according to the infant protocol Interfant were negatively associated with CME 9 (rho = −0.26, FDR < 0.001, Fig. 2E). Molecular BCP-ALL subtypes were strongly correlated with CME 1 and 2, underscoring the potential of the first CMEs to capture known biology. Specifically, high hyperdiploidy (HeH, rho = 0.77, FDR < 0.001) and *ETV6::RUNX1* (rho = −0.55, FRD < 0.001) were correlated with CME 1, and *ETV6::RUNX1* (rho = −0.73, FRD < 0.001) with CME 2 (Fig. 2F, Supplementary Data S3).

While CMEs 1, 2, 4, and 9 showed significant associations with known clinical parameters, including molecular subtype, age, treatment protocol, and risk group, the remaining CMEs exhibited more subtle and complex interactions across modalities and clinical features, warranting further exploration.

**CMEs are enriched for cellular functions and regulatory networks**

To better understand the molecular variables contributing to the CMEs, we first focused specifically on the top-ranked features (absolute weight > 0.6) in the DNAm and GEX datasets (Supplementary Data S4). This resulted in 1040 genes and 2681 CpG sites across the ten CMEs (Supplementary Data S5, S6). The CpG sites were annotated to genes using the GREAT annotation tool[46], resulting in 1027 genes for functional enrichment analysis (Supplementary Data S6). Of these, 64 top-ranked CpG sites were annotated to 36 top-ranked genes within the same CME, indicating a possible relationship between methylation level and transcriptional activity for these genes. Notably, this set included genes well-known to distinguish ALL subtypes and/or are involved in B-cell development: *BIRC7, DSC3, IGF2BP1, TCFL5, S100A16, PCDH9, IRF8, HPS4* and *PLVAP*[37,48,49,51] (Supplementary Data S7).

To identify biological pathways enriched within each CME, we performed ORA using the CME-associated genes. Because the overlap

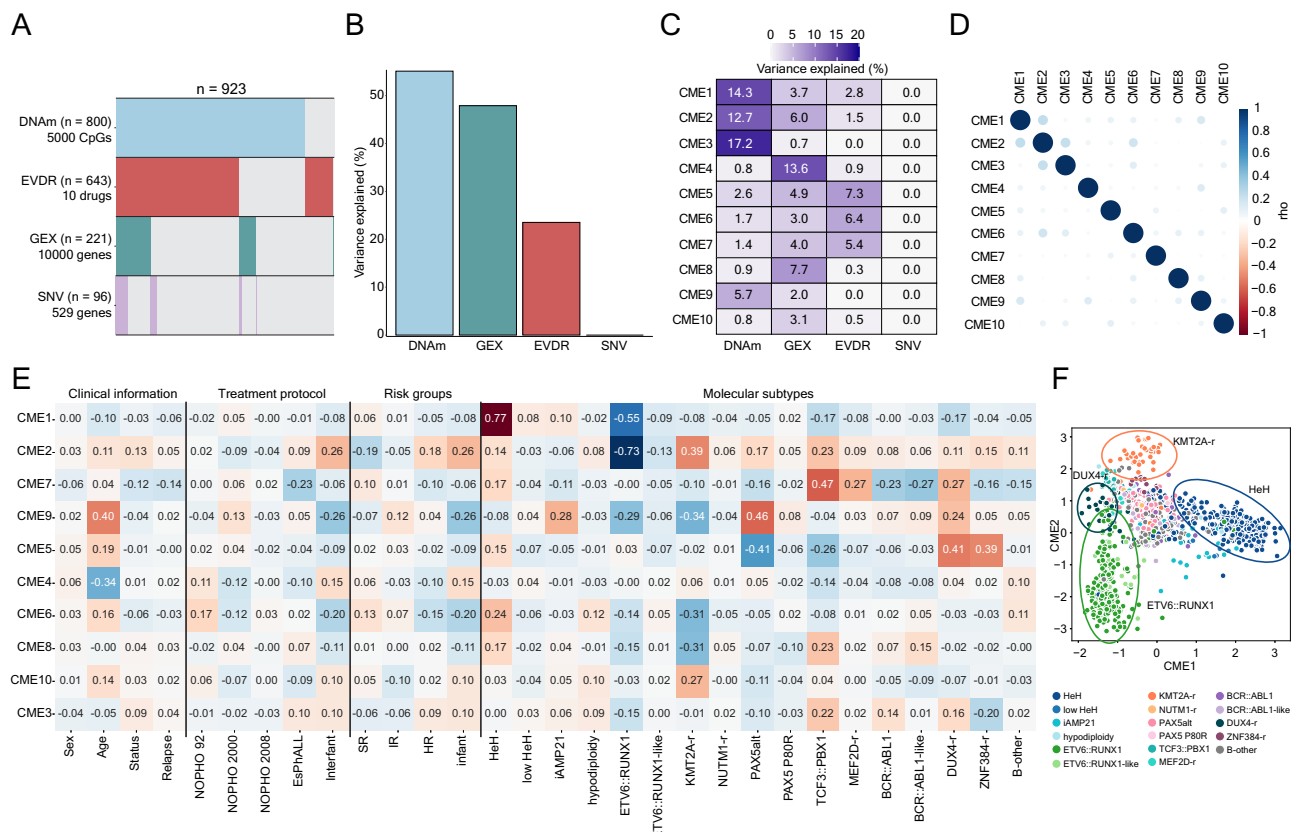

**Fig. 2 | Multi-Omics Factor analysis (MOFA) across 923 BCP-ALL patients.**
**A** Patient distribution per modality in the training dataset ($n = 923$). Missing data are indicated by light gray color. **B** Percentage of variance explained ($R^2$) by all cross-modal elements (CMEs) per data modality. **C** $R^2$ by CME for each data modality. **D** Pairwise correlation between CMEs (Pearson's correlation coefficient, rho). **E** CME correlation (rho) with clinical variables (i.e., sex, age, status (alive vs. deceased), and relapse (no relapse vs. relapse)), NOPHO treatment protocols, clinical risk groups (SR: standard risk, IR: intermediate risk, HR: high risk), and molecular subtypes. **F** Scatterplot showing the association between CME 1 (x-axis) and CME 2 (y-axis), color-coded by molecular subtype. Subtypes with defined boundaries are highlighted with ellipses.

between genes and annotated CpG sites was minimal, ORA was performed separately for each modality. This analysis revealed enrichment in 195 pathways and seven curated gene lists built based on prior knowledge[37,48–51] (FDR < 0.01, Supplementary Data S8). Next, we built pathway networks based on the degree of similarity (overlap coefficient > 0) between pathways by CME. Pathways with overlapping genes (>1 genes) were merged into a single "pathway cluster" and pathways without overlapping genes remained unclustered (Supplementary Data S8). To annotate each "pathway cluster", we tokenized each pathway into words using CountVectorizer, retaining the 20 most frequent terms. This text mining approach resulted in ten distinct annotations: *molecular ALL subtypes, molecular ALL subtypes including B-cell development, cell cycle regulation, cell cycle regulation including B-cell development, immune response, metabolic pathways, signaling pathways, T-cell pathways, transcriptional regulation,* and *mRNA processing* (Figs. 3A, S2, Supplementary Data S8). Corroborating our previous observations, the genes contributing to CMEs 1 and 2 were significantly enriched to *molecular ALL subtypes* (Fig. S2). CME 2, which explained 12.7% of the variance in the DNAm modality, was also enriched for *immune response* (Fig. 3B). CME 6, which accounted for 6.4% of the explained variance in the EVDR modality, was enriched for *cell cycle regulation* (Fig. 3C). CME 8, contributing to 7.7% of the variance captured in the GEX data modality, was enriched in *cell cycle regulation, metabolic pathways,* and *mRNA processing* (Fig. 3D).

### Inter-modal interactions within CMEs
To further explore inter-modal interactions within CMEs, we analyzed GEX, DNAm, and EVDR data from the training ($n = 923$) and test sets ($n = 308$)

separately, focusing on the top-ranked genes, CpG sites, and drugs (Supplementary Data S6, S7 and S9). For each CME, we constructed networks independently for positive and negative weights, which were organized into "communities" comprising clusters of related features with central "hub" features. Community size ranged from 2–10 clusters, and the number of hubs per CME ranged from 1–16 (Table 1, Supplementary Data S10, S11). Interactions across the three data modalities were observed in both positive and negative weight networks (Fig. S3, S4, Supplementary Data S10, S11).

For example, CME 2, which strongly correlated with ALL molecular subtypes (Fig. 2E), contained two DNAm communities with etoposide and doxorubicin as hubs (Fig. 4A). Unsupervised hierarchical clustering of DNAm from patients in the training set ($n = 800$), revealed two clusters of low and high DNAm levels (Fig. 4B). Increased DNAm levels were associated with higher survival indexes to both drugs (Mann-Whitney U test p-value < 0.001, Fig. 4C). Key CpG sites were annotated to ALL subtype-specific genes *IGF2BP1* ($n = 1$), *TCFL5* ($n = 2$), *TMED6* ($n = 1$), *BIRC7* ($n = 2$), and *DLGAP2* ($n = 1$)[37], and to the immune response gene *CSF2RB* ($n = 1$). Notably, the expression of *IGF2BP1*, *TCFL5*, *TMED6*, and *BIRC7* were also among the most influential features associated with CME 2 (Supplementary Data S7). Subtype distribution analysis of the low and high methylation clusters for both hubs revealed distinct patterns (Fig. 4D). The low-methylation etoposide-associated cluster was predominantly composed of *ETV6::RUNX1*-positive patients (83.9%, $n = 188$), indicating that this subtype is closely associated with the methylation status of the CpG sites in the etoposide hub.

In contrast, the doxorubicin hub spanned ALL subtypes, but with distinct subtype distribution. The low-methylation group was enriched for subtypes with generally favorable prognosis, including *ETV6::RUNX1*

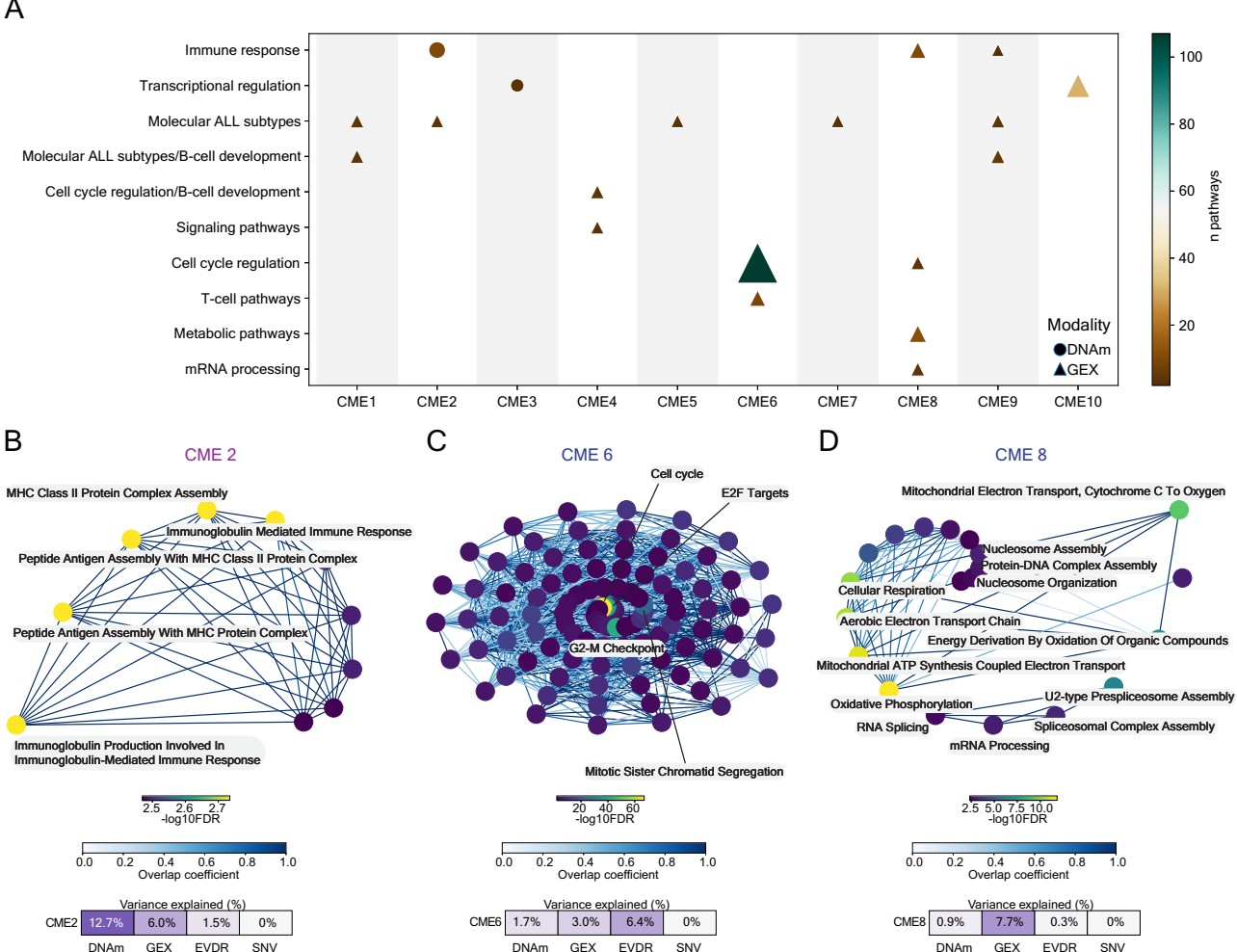

**Fig. 3 | Biological pathways associated with CMEs. A** Significantly enriched pathways (FDR < 0.01, y-axis) in features with negative or positive weights by CME (x-axis). The pathway clusters enriched in DNAm are denoted with circles and GEX with triangles. **B** Network of the *immune response* cluster for CpG sites with positive weights in CME 2. **C** Network of the *cell cycle* cluster for genes with negative weights in CME 6. **D** Network of the combined *cell cycle, metabolic pathways, mRNA processing* clusters for genes with negative weights for CME 8. The pathway nodes are color-coded based on -log₁₀FDR score, and edge connectivity is color-coded based on the overlap coefficient.

(82.1%, $n = 184$), *ETV6::RUNX1*-like (66.7%, $n = 10$), high hyperdiploidy (HeH, 69.2%, $n = 166$), and low HeH (100%, $n = 6$). In comparison, subtypes associated with poorer outcomes, such as *KMT2A*-r (100%, $n = 39$), *BCR::ABL1* (77.8%, $n = 14$), and *BCR::ABL1*-like (85.7%, $n = 18$), predominantly clustered within the high-methylation group. Surprisingly, the high-methylation cluster, associated with higher SI% after doxorubicin exposure, also contained about one third of HeH patients (30.8%, $n = 74$). For these patients, all CpG sites defining the hub ($n = 17$) were differentially methylated between the low- and high-methylation clusters, with 11 sites demonstrating > 20% methylation differences (Fig. 4E). Kaplan-Meier analysis confirmed a significant difference in relapse-free survival (RFS) between these HeH subclusters (log-rank $p$-value = 0.041, Fig. 4F). Specifically, HeH patients in the high-methylation doxorubicin-associated cluster exhibited both higher SI% and inferior survival outcome, an effect that remained significant after adjustment for risk group and treatment protocol (Wald test $p$-value = 0.039, HR = 2.34, 95% CI 1.04–5.2, Fig. 4F). Importantly, these patients did not differ in baseline characteristics such as age, white blood cell count (Mann–Whitney U test $p$-value > 0.05), or MRD response at day 29 ($n = 78$, Mann–Whitney U test $p$-value > 0.05).

While HeH is typically associated with a favorable prognosis, our findings align with previous reports describing a subset of high-risk HeH patients with poor outcomes[57,58]. Using the karyotype-based risk stratification proposed by Enshaei et al.[59], we classified HeH patients into good-risk

(+17 and +18, or +17 or +18 without +5 or + 20) and poor-risk (all other karyotype patterns). We did not observe a significant difference in the proportion of good- and poor-risk patients between the low- and high-doxorubicin response clusters (Fisher's exact test, $p$-value = 0.22). Furthermore, differential gene expression analysis comparing the high- ($n = 8$) and low-doxorubicin ($n = 24$) response HeH subclusters identified three differentially expressed genes, *CLIP4* (log₂ Fold Change (log₂FC) 1.14, FDR 0.035), *ABCB1* (log₂FC 1.2, FDR 0.049), and *CYBB* (log₂FC −2.64, FDR 0.043).

In a second example, in CME 8, low RNA expression of six histone genes (*H3C2, H2AC12, H2AC17, H2BC17, H2AC14* and *H2BC13*) was associated with higher SI% after exposure to vincristine (Fig. S5). Stratifying patients into three groups based on the expression levels of these genes revealed significant differences in SI% after ex vivo vincristine exposure, with the low-expression group showing higher SI% (Dunn's test, BH-adjusted $p$-value < 0.01, Fig. S5). In contrast, in CME 10, doxorubicin response was central and was associated with the expression of *FOSL1, FOSB, JUNB, TRIB1, LMNA*, and *ZFP36* (Fig. S5, Supplementary Data S10). Stratification of patients into three clusters based on the expression of these genes revealed significantly lower SI% after ex vivo doxorubicin exposure in the low-expression group (Dunn's test BH $p$-value < 0.001, Fig. S5).

Finally, we validated the inter-modal interactions using the test dataset ($n = 308$ patients), achieving 28.6–100% concordance for CME-specific

**Table 1 | Hub features per CME for positive and negative weights**

| CMEs | Positive CMEs Hubs | Negative CMEs Hubs |
|---|---|---|
| CME1 | cg00593243(*DUSP1*); *ITPRIPL2*; cg13280914; *CAMK1*; *RBM47*; cg01578875; *IL6R* | *CXXC5*; *KCNQ5*; *BEST3*; *CCNJL*; *RHOBTB1* |
| CME2 | Etoposide; Doxorubicin | *ARHGEF4*; *HAP1*; *EPHA7*; cg03717315(*SEMA4F*); *KCNN1*; cg16206460(*PGLYRP2*;*RASAL3*); *BIRC7*; *PCLO*; *LOXHD1*; cg26218983(*ZC3H12C*); cg07372034; *IGF2BP1*; cg16016176(*ADPRH*); *CLIC5*; *DSC3*; cg02851793(*SRGN*) |
| CME3 | Asparaginase; Dexamethasone | *FBXO41*; *FLT3* |
| CME4 | Dexamethasone; *ACSL1*; cg12150931(*ZNF385A*); Vincristine; *GABARAPL1*; *BHLHE40*; cg18425731(*ERMN*) | *VPREB1*; cg27392771; cg20752878(*ASGR1*); *CYTL1*; cg02225720(*HASPIN*); *RAG1*; cg20559385; cg06958535(*LAX1*) |
| CME5 | cg12150931(*ZNF385A*); *ATP9A*; cg12761788(*ENDOU*); *SALL4*; *DAPK1*; *GATA3*; *TTC28*; *CXCL2* | *LRRC14B*; cg27022853; cg05569131(*RAB44*); *CAMK2D*; *PHLDB2*; *TTN*; *LEF1* |
| CME6 | cg06452129; *GPRIN3*; cg16997486; *GIMAP5*; *TRAT1*; *TC2N*; cg16977751; *LTB*; *GIMAP7*; cg24394336 | cg08097359 |
| CME7 | cg17183174(*COMMD6*); cg15354065(*DGKA*); *LDLRAD3*; *GLDC*; cg00524374 | *AHR*; *ANTXR2*; cg14429979; cg00915974; cg11952493(*MPP7*); cg16783349; *SAMSN1*; cg08015762 |
| CME8 | *CIITA* | cg07311994; Vincristine; *LGALS1*; *RGS2*; *GPR183*; cg04450037 |
| CME9 | cg24618492(EPN2); cg25789861(DSC3); *MAPKBP1*; cg09144073(*STXBP5*); cg02312409(*RNF217*); cg19478500(*RNF217*); cg19832521(*NOVA1*); *CD9*; cg03017520(*DSC3*); *GALNT2*; cg13434989(*EDNRB*); *NIBAN3*; *PKIG* | cg16747164; cg05754179; cg08478016; *KCNN1*; *DSC2*; cg01984854; *BASP1*; *IGF2BP1*; *PLCB4*; *DSC3*; *PRKAR2B* |
| CME10 | cg17604985 | Amsacrine; Doxorubicin |

A hub is defined as a feature with > 3 connections to features that belong to another data modality.

hubs (Supplementary Data S12, S13). Specifically, we confirmed associations within CME 2 (etoposide, 42/70 CpG sites; doxorubicin, 5/17 CpG sites), CME 8 (vincristine, *H3C2* and *H2BC13*), and CME 10 (doxorubicin, *FOSB*, *JUNB* and *ZFP36*).

### Functional annotation of CMEs
Next, we integrated our key findings and summarized the interactions, mechanisms, and pathways associated with each CME (Fig. 5). This map provides a unified and comprehensive view of CME-driven associations. Key annotations related to BCP-ALL subtypes or B-cell development were linked to several CMEs (1, 2, 4, 5, 7, and 9), underscoring their central roles in the disease. Immune-related responses were also prominent across CMEs 2, 6, 8, and 9, encompassing diverse mechanisms such as Major Histocompatibility Complex (MHC) class responses (CME 2), T-cell pathways (CME 6), humoral responses (CME 8), and interleukin-mediated responses (CME 9). Ex vivo drug responses were mapped to several CMEs (2, 3, 4, 6, 8, and 10). Notably, CME 4 captured variability in glucocorticoid (dexamethasone and prednisolone), thioguanine, and vincristine responses across GEX and DNAm data (Fig. S3). Cell cycle regulation emerged as a central driver across CMEs 4, 6, and 8. CME 8, in particular, was characterized by regulation of metabolic pathways and histone-mediated vincristine response.

Distinct processes were identified for specific CMEs, such as CME 10, which was uniquely associated with cell proliferation and apoptosis. This CME featured AP-1 complex genes (*FOSL1*, *FOSB*, and *JUNB*), which were positively linked to response to topoisomerase inhibitors, such as doxorubicin and amsacrine.

### CMEs reveal the prognostic value of ex vivo drug responses
To evaluate the prognostic utility of CMEs beyond baseline clinical risk groups, we implemented gradient boosted survival models and evaluated their performance using the CV c-index[54] across the 3 × 5 repeated stratified folds, as well as the c-index on the independent test dataset. For models incorporating molecular and drug response data, we applied the MOFA imputation step to generate complete feature sets on the complete training set, and the training was performed using the top features (absolute weight > 0.6).

Pairwise CV score comparisons between the clinical baseline model and 82 CME-based models identified three models with significantly improved performance (FDR < 0.05, Supplementary Data S14, Fig. 6A). In all three cases, combining CME features (i.e., ex vivo drug responses) with clinical data improved prognostic accuracy. Specifically, the top performing models derived from CME 2 (FDR 0.01, mean CV c-index 0.64, 95% CI 0.62–0.66), CME 7 (FDR 0.02, mean CV c-index 0.64, 95% CI 0.62–0.66), and CME 8 (FDR 0.005, CV c-index 0.62, 95% CI 0.60–0.65), all outperforming the baseline clinical model (mean CV c-index 0.59, 95% CI 0.56–0.61) and generalized well to the test dataset (Fig. 6B, Supplementary Data S14). Specifically, these models achieved higher c-index (CME 2: 0.663, CME 7: 0.667, and CME 8: 0.714) compared to the clinical model alone (0.656) on the test dataset.

Top drugs contributing to CME 2 were cytarabine, etoposide, and doxorubicin. Notably, in the previous section, we linked ex vivo response to doxorubicin to the DNA methylation levels of 17 CpG sites, which further stratified patients by relapse-free survival (Fig. 4). CME 7 is one of the intermodal CMEs, spanning across all data modalities, yet the top features of this CME were ex vivo response to dexamethasone, prednisolone, doxorubicin, etoposide, thioguanine, and vincristine, explaining most of the variance in the EVDR data (5.4%). Finally, CME 8 was primarily driven by vincristine, L-asparaginase, and cytarabine. Of particular interest, vincristine SI% was negatively associated with the expression levels of six histone genes (Fig. 4).

## Discussion
Risk stratification for pediatric BCP-ALL remains imperfect, reflecting both the biological diversity of the disease and our incomplete understanding of its molecular drivers. To address this challenge, we integrated DNA methylation, transcriptional, and drug response data from 1,231 patients to construct an integrated molecular map of pediatric BCP-ALL. This analysis identified ten distinct CMEs that captured key aspects of leukemia biology and functional drug responses.

Although our integrative CMEf and CMEf-all models did not generally outperform the clinical baseline, our framework proved effective for uncovering informative feature combinations. Our data demonstrated that survival models incorporating CME-derived drug response features (CMEs 2, 7, and 8) yielded added prognostic value. In this context, integrative multi-omics analyses primarily supported feature prioritization rather than directly enhancing the predictive performance of survival models.

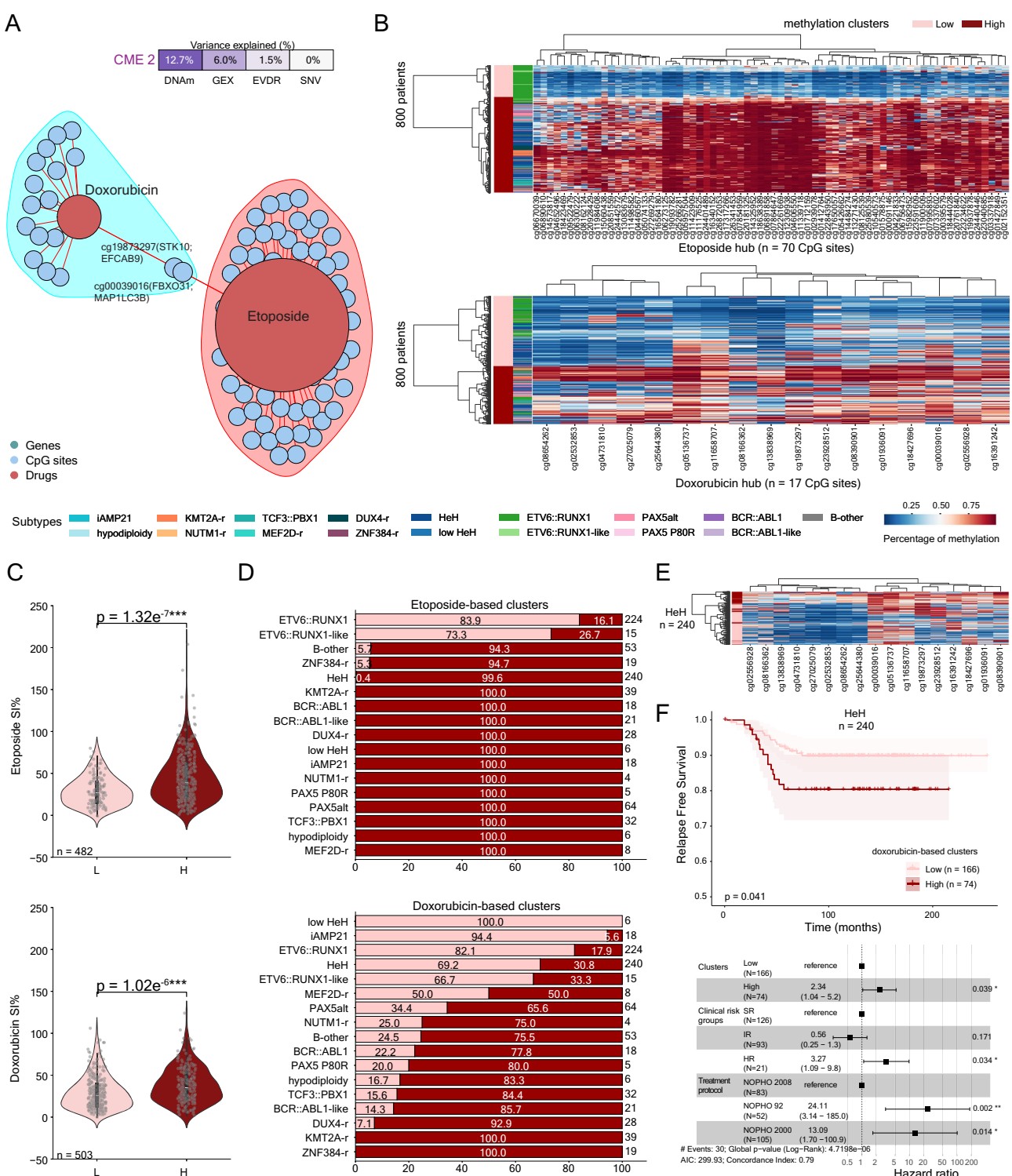

From a pathophysiological perspective, our data highlighted that CMEs 1 and 2 were significantly correlated with the molecular subtype. This is not unexpected, as gene expression and DNA methylation biomarkers are driven by the known, subtype-specific heterogeneity of BCP-ALL[1,2,10,60]. However, in the inter-modal network analysis for CME 2, we noted an unexpectedly high proportion of patients with low-risk molecular subtypes (e.g., HeH) stratified, using DNA-methylation data, in a subgroup of patients with higher SI% when exposed to topoisomerase inhibitors (i.e., doxorubicin). These patients exhibited significantly inferior outcomes, independent of other clinical variables. Differential

gene expression analysis between the two HeH subgroups revealed upregulation of *CLIP4* and *ABCB1* and downregulation of *CYBB* in patients with a hypermethylation pattern and high doxorubicin SI%. *ABCB1*, in particular, which encodes the efflux pump P-glycoprotein, is a key drug-efflux transporter and well-known driver of anthracycline resistance[61], suggesting a drug-refractory cellular state in this group. These results further support the presence of molecular subgroups within the HeH subtype that exhibit differential clinical outcomes, consistent with previous studies[57–59,62]. Importantly, this study demonstrates that integrative multiomics analysis can uncover previously overlooked

**Fig. 4 | Inter-modal networks and correlations in the train dataset (**$n = 923$ **samples). A** Inter-modal networks for features with positive weights and Pearson's absolute correlation coefficient (rho) > 0.2 in CME 2. The variance explained by CME 2 is shown at the top of the panel. Positive correlations are denoted with red. The size of each correlation (edge) is based on the correlation value, while the size of each circle (vertex) represents the number of connections + a weight of 10. **B** Heatmaps of the DNA methylation levels for CpG sites (*x*-axis) across 800 patient samples (y-axis) mapped to the etoposide (top) and doxorubicin hubs (bottom) ordered by unsupervised hierarchical clustering. The methylation clusters and the molecular subtypes are shown as annotation bars on the y-axis. **C** Violin plots of the distribution of surviving cells (SI%, y-axis) after etoposide (top) and doxorubicin (bottom) treatment, colored by DNA methylation cluster group (low-high, x-axis). Two-sided Mann-Whitney U test *p*-values: *** < 0.001, ** < 0.01, * < 0.05, ns: non-significant. L: low, H: high. **D** Molecular subtype distribution in the low and high

methylation clusters across 800 patients with DNAm data available on the training dataset in the etoposide hub-based clusters (top) and in the doxorubicin hub-based clusters (bottom). **E** Heatmap of the DNA methylation levels for CpG sites (x-axis) across 240 patient samples with high hyperdiploid (HeH) subtype (y-axis) ordered by unsupervised hierarchical clustering and color-coded by the low- and high-doxorubicin-based methylation clusters. **F** Kaplan-Meier survival curves for the doxorubicin hub-based clusters, illustrating relapse-free survival (RFS, y-axis) over time (x-axis) for HeH patients ($n = 240$) with available DNAm data (top). The log-rank test p-value access the difference between the two clusters and is denoted on the bottom left corner. The error bands define the upper and lower 95% confidence intervals. Cox proportional hazards regression adjusted for treatment protocol and clinical risk group (bottom). The forest plot shows the hazard ratio with the corresponding 95% confidence intervals for relapse in the doxorubicin hub-based clusters.

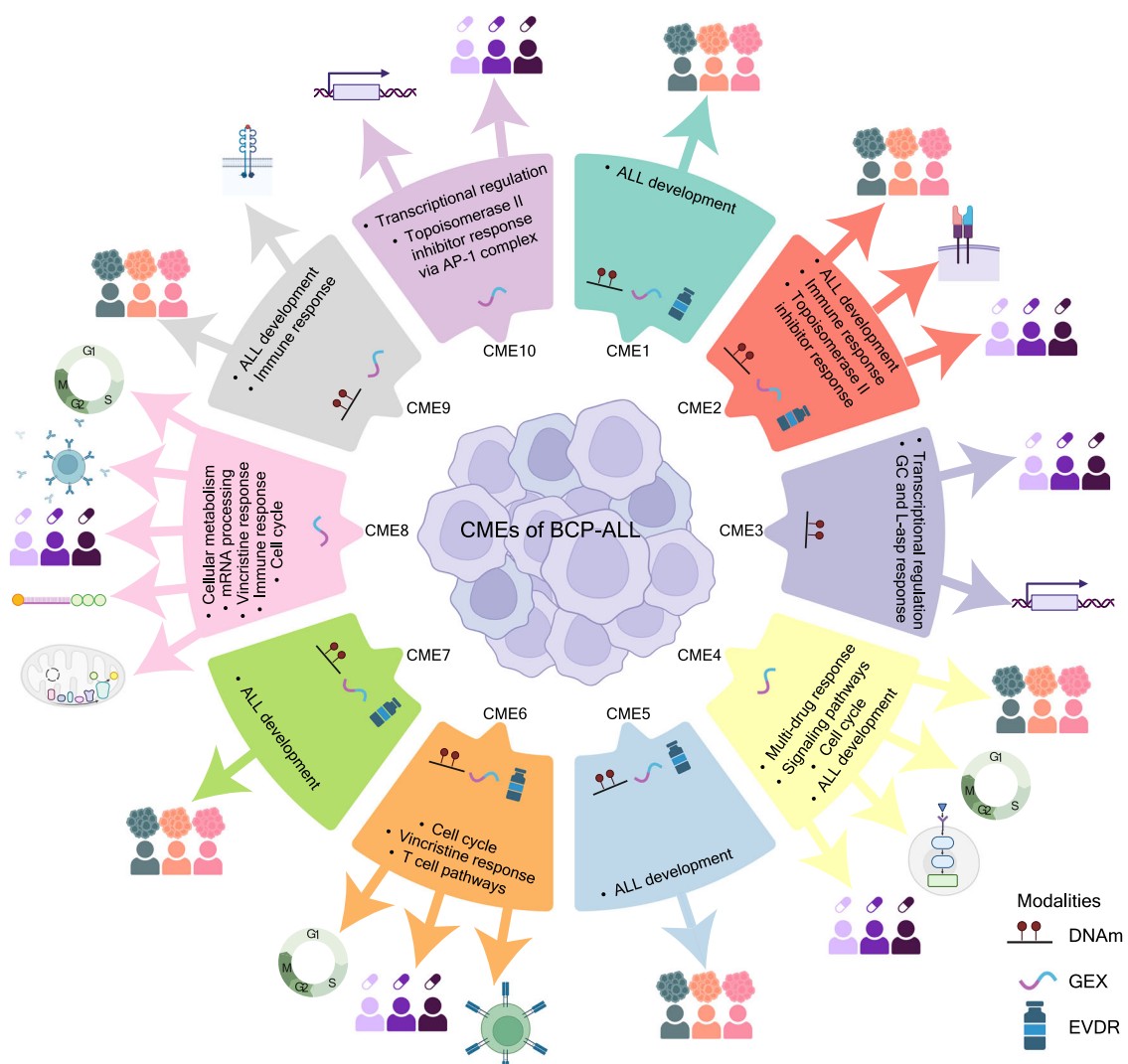

**Fig. 5 | Graphical functional annotation map across CMEs.** The data modalities with >1% variance explained by MOFA are highlighted in the center of each CME. CG: glucocorticoids, L-asp: L-asparaginase, DNAm: DNA methylation, GEX: Gene

expression, EVDR: ex vivo drug response. The figure was created in Adobe Illustrator, but the icons were adapted from Biorender.com.

molecular interactions influencing doxorubicin response. In line with our observations, Lee et al.[30] reported variable responses to anthracyclines among HeH patients, where daunorubicin-resistant cases demonstrated sensitivity to other compounds such as trametinib, venetoclax, and ibrutinib, suggesting potential alternative treatment options.

Furthermore, in the inter-modal network analysis of CME 2, of the 70 hub-specific CpG sites in the etoposide hub, only ten overlapped with the ALLIUM[37] ALL subtype differentiating CpGs, while none were found on the doxorubicin hub. This finding suggests that drug responses may be influenced by other factors beyond the molecular subtypes, including

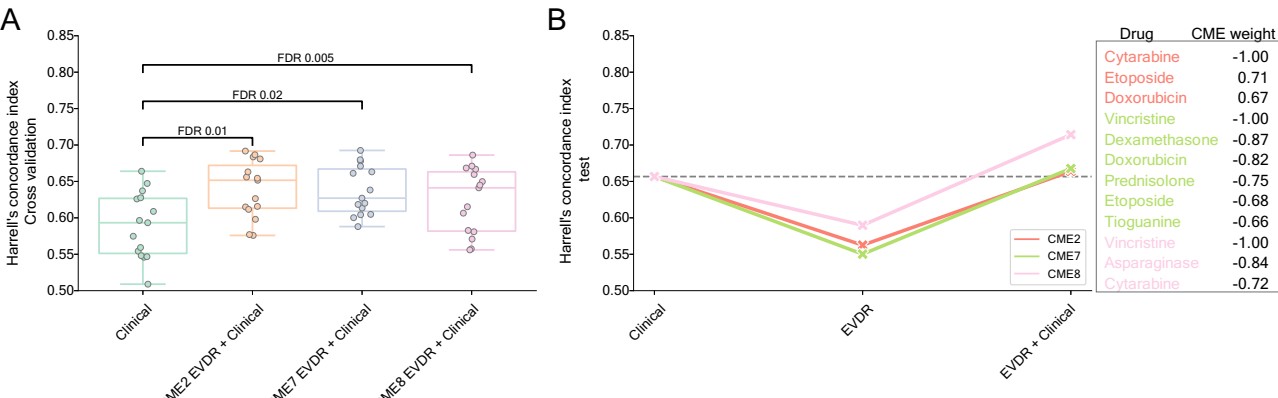

**Fig. 6 | Survival model performance evaluation. A** Harrell's concordance index (c-index, y-axis) for each model (x-axis) across individual CMEs on the 3×5 repeated stratified cross-validated sets. The whiskers demonstrate the distribution of the data points beyond the lower and upper quartiles. Two-sided Wilcoxon signed-rank test Benjamini-Hochberg (BH) adjusted *p*-values are shown above each boxplot. **B** c-index (y-axis) for the clinical baseline model, the CME-derived EVDR models, and combined models (x-axis) on the test dataset (*n* = 308). The grey dotted line represents the c-index (0.657) for the clinical baseline model. The top drugs color-coded by CME and their impact (weights) on the respective CMEs are provided as a figure legend.

epigenetic modification, given the exclusive connection of the hubs to CpG sites.

Cell cycle regulation plays a dual role in chemotherapy, either acting synergistically with or counteracting drug efficacy[63]. The drugs used in this study exhibit diverse mechanisms of action, with some targeting all cell cycle stages (cell cycle non-specific agents) and others affecting only actively dividing cells (cell cycle-specific agents). For instance, alkylating agents, anthracyclines, and topoisomerase inhibitors like doxorubicin, amsacrine, and etoposide are non-specific agents, effective against cells irrespective of their cell cycle phase[63]. In contrast, antimetabolites (e.g., cytarabine, thioguanine) and vinca alkaloids (e.g., vincristine) are cell cycle-specific, limiting their effects to actively dividing cells[63,64]. Here, cell cycle regulation emerged as a main feature of CMEs 6 and 8, where cell cycle-dependent drugs served as central hubs. Notably, in CME 8, higher expression of genes encoding histone proteins (markers of proliferating cells) was associated with lower SI % following vincristine exposure. Our findings underscore the role of core histone protein expression in mediating drug sensitivity in ALL. These results align with prior studies showing that slow or non-cycling cancer cells are associated with therapy resistance[65]. For example, ALL samples with higher cell proliferation rates showed distinct sensitivity profiles to cell cycle-related drugs in ex vivo xenografts models[66] and cell cycle checkpoint regulation is critical in ALL treatment response[67]. Potential strategies include combining vincristine with agents active in quiescent cells, such as BCL2 inhibitors (venetoclax)[68]. CME 6 and 8, along with their combined survival models, did not outperform the clinical model, suggesting that while cell-cycle-specific features are biologically relevant, they may not capture the full variability required to enhance model performance. However, the EVDR-combined survival model for CME 8 outperformed the baseline model, suggesting a link between ex vivo drug responses and clinical outcomes.

Alterations in cellular metabolism, oxidative phosphorylation, and cellular respiration pathways are established mechanisms of chemoresistance and potential therapeutic targets in acute leukemia[69]. CME 8 was enriched to metabolic and cell cycle-related pathways, showcasing evidence of the relationship between metabolism and cell cycle, as previously described[70]. Deciphering this bidirectional relationship may allow the use of drugs targeting metabolic pathways, which can in turn affect cell cycle progression[70,71].

The top-ranked genes in CME 10 were enriched in transcriptional regulation pathways, including TNF-alpha signaling via NF-kB, p53, apoptosis, and hypoxia. Notably, the expression levels of *FOSL1*, *FOSB*, *JUNB*, *TRIB1*, *LMNA* and *ZFP36* were positively correlated with response to doxorubicin. These genes are components or regulators of the Activator Protein 1 (AP-1) transcriptional complex[72–76]. AP-1 is known to play a

central role in transcriptional regulation, cell proliferation, apoptosis, and metastasis. Our findings suggest that patients with high AP-1 gene expression may exhibit reduced suitability for treatment with topoisomerase inhibitors like doxorubicin. Although, topoisomerase inhibitors and glucocorticoids are mechanistically distinct, previous studies showed that JUN knockdown mediates resistance to glucocorticoids in T-ALL[73], highlighting the broader role of AP-1 components as potential therapeutic targets in ALL.

Multi-Omics integration approaches, such as MOFA[22], are powerful tools for vertically integrating and reducing the dimensionality of complex multimodal datasets. They employ various strategies, including probabilistic, multivariate, network-based, similarity-based, fusion, or correlation-based methods[33]. The number of available tools continues to grow, yet systematic benchmarking remains limited. Although benchmarking MOFA against other integration tools was beyond the scope of this study, our choice was guided by the need for an unsupervised method capable of handling partially overlapping data, while remaining computationally efficient and interpretable. Despite the limitations of using a linear approach such as MOFA, interpreting the results within a biological context remains essential.

MOFA provided a holistic exploration of multimodal datasets, while also identifying individual modalities associated with specific biological aspects. In addition, as the authors of MOFA demonstrated by masking the data, the method effectively handles missing data (within or between assays) through imputation[22]. This mitigated the risk of reduced sample size due to modality imbalance and maximized the statistical power of our survival analyses. By incorporating all available patient samples (923 vs. 147 in the train set and 308 instead of 49 in the test set), we were able to build robust survival models, a crucial step for developing reliable, predictive tools, which can generalize effectively across diverse cohorts. However, when the proportion of missing samples is high (e.g., GEX–76%), the imputation step may not be effective for this modality. Therefore, we recommend integrating all modalities to mitigate the effect of missing data. One limitation of MOFA is the lack of a method for projecting CMEs onto external datasets. To address this, we used MOFA as a proxy to identify the most influential features (genes, CpG sites, and drugs) for downstream analyses, which were validated on the test data. In the context of other diseases, Iperi et al.[77] and Pekayvaz et al.[78], and for healthy blood profiling, de Visser et al.[79] successfully applied MOFA and validated their findings in other datasets, focusing on either multiple or single data modalities. Finally, given the associations between drugs and GEX profiles in CMEs 8 and 10, exploring these relationships at the proteomic level may yield additional insights given the significant role of the proteome in reflecting the impact of mutations in ALL[13] and its potential as a source of druggable proteins[11].

As our study is based on a retrospective cohort, the long follow-up time is a major strength. However, treatment protocols have evolved over time. End-of-induction (EOI) minimal residual disease (MRD) assessment, now routinely used for risk stratification, was only available for 348 of the patients in our dataset. The limited number of patients stratified by MRD data prevented its broader inclusion in our analyses, as selective incorporation could introduce bias. However, we did not observe any association between CMEs and the main treatment protocols (NOPHO ALL-92, −2000, or −2008) except for infants under one year of age treated according to the Interfant protocol. This distinction aligns with the unique molecular and age-based stratification criteria of Interfant[35], which were expected to be reflected in the CME data. The patients included in this study also preceded the implementation of copy number alteration-based stratification[80]. While our findings remain relevant for understanding leukemia biology and treatment responses, future studies should strive to fully integrate MRD and copy number analyses to align with contemporary risk stratification strategies.

## Conclusion

Our multiomics integration study revealed how individual and combined data modalities contribute to key biological processes and clinical outcomes in pediatric BCP-ALL. This study underscores the potential of multiomics profiling to improve risk estimation and disease stratification in BCP-ALL. By reducing the complexity of large datasets to a few clinically significant factors, we identified overarching molecular signatures with prognostic relevance. These findings highlight the value of integrative, factor-based approaches for advancing clinical, basic, and translational research in BCP-ALL.

## Data availability

The GEX count matrix was retrieved from GSE227832. DNA methylation data were available from their original studies, under controlled access via GSE49031, https://doi.org/10.17044/scilifelab.22303531[81] and https://doi.org/10.17044/scilifelab.26096371[82] (https://figshare.scilifelab.se/). The processed EVDR data are available in Supplementary Data S15. Source data underlying the analyses in the main Figures are available in Supplementary Data S16. The raw FASTQ and IDAT files are not shared publicly due to confidentiality and ethical restrictions. Any data inquiries may be submitted to the corresponding author and will be reviewed within four weeks. Access may be granted to qualified researchers subject to a data use agreement restricting use of the data to approved research purposes, prohibiting attempts to re-identify participants, and requiring compliance with applicable ethical and data protection regulations.

## Code availability

All R and Python scripts, and the environment requirements to reproduce the analyses, are openly available at our GitHub repository https://github.com/Molmed/Krali_2026_MultiOmics[83]. All figures can be reproduced using the provided Supplementary Data files and the workflow scripts.

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

## Acknowledgements

This work was supported by grants from the Swedish Research Council (2019-01976 to JN), the Swedish Cancer Society (CAN2022-2395 to JN), and the Swedish Childhood Cancer Foundation (PR2022-0082 and HFT2023-0011 to JN, TJ2020-0039 to AH). We thank the SciLifeLab National Genomics Infrastructure, SNP&SEQ Technology Platform, which is funded by the Swedish Research Council and the Knut and Alice Wallenberg Foundation, for assistance with data generation. The data handling was enabled by resources provided by the Swedish National Infrastructure for Computing (SNIC) and the National Academic Infrastructure for Super-computing in Sweden (NAISS). SNIC and NAISS are partially funded by the Swedish Research Council through grant agreement no. 2022-06725. We especially thank the ALL patients who contributed samples to this study.

## Author contributions

J.N. and O.K. conceived the study. O.K., J.S., and A.L. analyzed the data. J.S. ran MOFA. O.K. performed downstream analyses and generated the figures and tables. GL and AH provided clinical material, data, and expertise. C.A. provided the FMCA data and expertise. JN provided funding. O.K., A.P.E., J.S., and J.N. wrote the manuscript. D.G. provided expertise in survival modelling and pathway analysis. T.E. and M.H. critically reviewed the manuscript and provided expertise in data integration. All authors read and approved the final version.

## Funding

## Competing interests

The authors declare no competing interests.
