## [Transparent Peer Review file · Communications Medicine]

An integrative molecular map of pediatric B-cell precursor acute lymphoblastic leukemia

Corresponding Author: Dr Jessica Nordlund

Version 0:

Reviewer comments:

Reviewer #1

(Remarks to the Author)

Krali et al. applied Multi-Omics Factor Analysis (MOFA) to integrate genomic, epigenomic, transcriptomic, and ex vivo drug response data from a large cohort of 1,231 pediatric patients with B-cell precursor acute lymphoblastic leukemia (BCP-ALL). This analysis identified ten key patterns, referred to as cross-modal elements (CMEs), which were subsequently used for pathway enrichment and intermodal network analyses. The impact of these CMEs on patient outcomes was assessed through survival modeling. Overall, the study offers valuable insights into multi-modal data integration and interpretation, with potential clinical relevance for patient stratification. The manuscript is well-structured, and the data are presented clearly and concisely.

Major comments:

A key concern is the moderate predictive performance of the proposed survival models. Baseline models using only clinical data achieved a concordance index (c-index) of 0.657 on the test set. The best-performing CMEf model reached a c-index of 0.699, which increased marginally to 0.703 when combined with clinical data (CMEf + Clinical). While these results suggest that integrating multi-modal data may improve outcome prediction, the conclusion would be more compelling if the authors demonstrated that the observed differences are significant. This could be addressed through stratified cross-validation, although it's important to recognize that CMEs may differ across iterations. Nevertheless, focusing on the best-performing model and the aggregated CME features from each iteration would allow for meaningful statistical testing.

Minor comments:

- 1) The title states "An integrative molecular map of pediatric B-cell leukemia" but reported is solely B-ALL.
- 2) The Results section of the Abstract lacks specificity and does not clearly summarize the key findings presented later in the manuscript. Including more concrete outcomes would improve clarity and better reflect the study's contributions. In Line 34: "This enabled the identification of molecular and drug dependencies relevant to patient risk stratification." Please state one key example at least. Lines 35-37: most important survival model results should be stated with numbers.
- 3) Line 121: On what basis were the drug concentrations selected?
- 4) Line 153: How were the top background CpG and genes defined? Based on their variance?
- 5) Line 225: Table S2 contains a mixture of category labels (TRUE, 1, FALSE, 0), please unify.
- 6) Line 252: CME 2 had later the best performance for survival prediction and in addition to ETV6::RUNX1 also KMT2A and infants are significantly correlated. This is worth stating in the results.
- 7) Line 271: "and pathways with without overlapping" please remove "with"
- 8) Line 292: "Increased DNA methylation levels were associated with higher survival indexes (indicative of resistance) to both drugs (Dunn's test BH p-value <0.001, Fig. 4B-C). " What was the method applied to defining L and H patients? If clustering was used, please add a dendrogram to the heatmaps in Figure 4.
- 9) Lines 299-311: The identification of a high-risk hyperdiploid subgroup is a particularly interesting finding and warrants

greater emphasis. It would be valuable to include this subgroup as an additional panel in Figure 4 to highlight its distinct characteristics. Additionally, please consider adding a subtype annotation column—at least for the most frequent subtypes—in the heatmaps presented in panels B and C. Furthermore, it would be helpful to clarify whether there are differences between the two hyperdiploid groups in terms of cohort distribution, patient age, clinical risk stratification, karyotypes, methylation patterns, or gene expression profiles. Lastly, are there any observed outcome differences between these groups in patients treated under more recent clinical protocols?

10) Line 352: “Combining CMEs with clinical data improved prognostic performance.” Is this significant?

11) Fig 4A. What value determines the icon size?

Reviewer #2

(Remarks to the Author)

In this manuscript, Krali et al describe an integrative analysis of pediatric B-cell precursor ALL, based on retrospective DNA methylation, gene expression, SNVs and drug sensitivity data combined with multi-omic factor analysis (MOFA). This is a presentation of a great dataset for integrative analyses. Perhaps the most interesting finding is the identification of a high risk group of patients based on DNA methylation within the high hyperdiploid ALL group, commonly known as good outcome subgroup. This subcluster of patients with high DNA methylation patterns show poorer response to doxorubicin and etoposide. The authors claim that their identification of so-called cross-modal elements and subsequent incorporation into survival models improved risk stratification. However, according to the data in the last figure 6, similar predictions in the test set were achieved between baseline models using clinical data only or such integrative CME analyses. Thus, it is unclear what the conclusion now really is, what should be done now in terms of molecular and functional analysis?

Next to this, one major point here is, what is the congruence and/or relation to current risk assessment strategies, in particular CNA-based analyses (Moorman et al., blood 2014), or MRD-based risk stratification as used in current protocols?

Other major points:

- In Fig 2E-H, association of CMEs with clinical parameters and genetic subtypes – what are the p-values here?
- In line 255, it is stated that CME 1, 2, 4 and 9 were associated with known clinical parameters, which are these, what are the data? In Fig. 2E it is unclear what status and relapse describe. What does the size of the correspond to circles encode (p-value? legend scale?). Consider using a uniform representation.
- Can the authors elaborate why SNVs and deletions in genes like IKZF1, Pax5 and TP53 do not contribute to the variance? Is Pax5 alteration status (CME5/9) fully captured by other modalities?
- The idea of a training and a test set would be to train one set and to analyze these results then in the test set – yet it is unclear throughout the manuscript what analysis was done in the training set and what in the test set, and what on the whole dataset.
- It is unclear what the survival indexes (Figure 4C) indicate, survival at what drug concentration, based on what parameter? What is the fraction of surviving cells relating to? Negative control (e.g. DMSO) at the endpoint (72h) or fraction of cells dying over time? What does ex vivo resistance to cytotoxic anthracyclines refer to? Is this a complete non-response of the cells? It seems that the boxplots differ only marginally. Is it justified to talk about ex vivo resistance here? Consider showing individual drug responses in the boxplot.
- Could the authors speculate- based on their integrative analysis, on novel drug targets potentially in different subgroups?
- CME3 is labelled as asparaginase and dexamethasone hub – two of the most relevant drugs of induction therapy. One would expect these to be highly predictive – yet this appears not the case, what is the explanation?
- In Figure 4E and H, the authors talk about low, intermediate and high expression levels, what does this mean, what difference in expression are we talking about here?
- It is not fully clear how the networks Figures in Fig3B-D should be interpreted. It is difficult to judge the number of connectivities between nodes. Consider indicating a scale for the connectivity strength of the edges.
- The conclusions as well as the title and the abstract of the manuscript are fairly generic. Please consider being more precise and concrete about the relevance for precision medicine. What would be actionable clinical conclusions?

Reviewer #3

(Remarks to the Author)

In this study, Kralli et al. combine SNV, gene expression, DNA methylation, and ex vivo drug response data through Multi-Omics Factor Analysis (MOFA) to characterize data from 1231 pediatric patients with BCP-ALL. This led to the identification of ten cross-modal elements (CMEs) explaining >2% of the variance in the training dataset. Each CME was thoroughly profiled, identifying key features and pathways. Functional modules for each CME were identified, within and across datasets, and gene expression and DNA methylation were associated with particular drug responses in CME-specific networks. Finally, individual CMEs were incorporated into survival models, some of which slightly outperformed a clinical baseline model.

This work represents a thorough multi-omic characterization of pediatric BCP-ALL. Using the CMEs, authors have identified valuable connections between DNA methylation and gene expression involved in cancer-related processes, as well as informative interactions of both of those modalities with drug response data. These findings identify novel relationships that (if functionally validated) may lead to significant improvements in patient stratification and treatment selection. One facet that was not thoroughly explored in this manuscript is the relationship of CMEs, key features of each CME, and molecular subtypes. Particularly since subtypes are used to stratify pediatric BCP-ALL patients and inform the use of targeted therapies, a subtype-focused exploration would be highly clinically relevant. Additionally, the incorporation of CMEs into survival models is of limited clinical utility. Since performing all (or in some cases any) of the assays used to identify CMEs is impractical at the bedside, the slight improvement in model performance with the addition of CMEs is an academic exercise. The survival modeling would be significantly more impactful if a handful of features (which could potentially be assayed clinically) were extracted from a CME of interest and their addition to the baseline clinical model improved performance.

Overall, the methodology in this work is sound and well-considered. Significant analysis has revealed novel interactions between gene expression, DNA methylation, and drug response. However, the lack of subtype-specific querying of the data and minimal clinical utility of the CME-based survival models limit the impact of this work on the characterization and treatment of patients with BCP-ALL.

Major points:

1. Since the authors observe strong correlation of CME1 and 2 with molecular subtype, it would be beneficial to show a (PCA-style) plot of patient data across CME1 and CME2 (similar to the MOFA publication (Argelaguet et al., 2018)'s Figure 2E), with patients colored according to molecular subtype.
2. In the section "CMEs are enriched for cellular functions and regulatory networks," the statement that "Of these [genes in table S6], 36 genes were shared within the same CME". As written, this sentence suggests that 36 of the 1027 genes were enriched in one particular CME. Only by exploring the supplementary data does it become clear that the genes and CpG sites associated with those genes are top ranked features within the same CME, which alters the significance of the section. The transition within the same paragraph to CME-specific pathway enrichment deepens this confusion.
3. In Figure S5A-B, the scales are unclear - does "percentage" represent the percentage of patients in the cluster of a particular subtype, or the percentage of patients with a particular subtype who fall into that cluster? A stacked bar plot could alleviate some of this confusion. This is especially significant because a large portion of the section "Inter-modal interactions within CMEs" is spent discussing low- vs. high-methylation groups of patients belonging to either ETV6::RUNX1 or HeH subtypes, but it's difficult to ascertain whether these observations are subtype-specific or not, and Figure S5 does not adequately clarify the question. Notably, both of these subtypes tend to have good outcomes, so further discussion of other subtypes would be appreciated. Clarity of this question in Figure 4 could be improved by adding a column to the left of the heatmaps (in panels B, E, and H) colored according to patient subtype, alongside the column colored according to L/H or L//H categorization.

Minor points:

1. Authors state in the MOFA section of Methods that "a subset of the most variable 5,000 CpG sites and 10,000 genes were used." How were these numbers of features selected? Considering the complexity of the disease and multiple subtypes, it's possible that a feature of interest may not have been included in the analysis. Were any features outside of this set evaluated or considered?
2. In the Network analysis section of Methods, how was the absolute weight >0.6 selected as the threshold for most impactful features? Do the biological insights fundamentally change with a cutoff of 0.5 or 0.7?
3. In figure 2E, which clinical variable does 'Status' represent? Also, what does the size of each dot represent?
4. In addition to pathway enrichment separately across data modalities for each CME, authors may consider an integrated approach such as that offered by ActivePathways (Paczkowska et al., 2020).
5. In the first paragraph of the Discussion, authors claim that "Survival models incorporating CME-based features demonstrated a significant improvement in predictive accuracy compared to traditional clinical risk stratification." Given the c-index of best models (0.707) compared to the baseline clinical model (0.657), some statistical support for the claim of "significant improvement" would be beneficial.

Version 1:

Reviewer comments:

Reviewer #1

(Remarks to the Author)

The authors have thoroughly revised the manuscript and successfully addressed all major concerns raised by the reviewers. The technical description of the ex vivo drug response profiling has been improved, enabling the reader to clearly

understand what (and how) drug concentrations were selected and responses evaluated. For the MOFA analysis, the authors have now included a detailed description of the initial feature selection steps and have explicitly stated throughout the manuscript which analyses were performed on the training versus test data.

A cross-validation approach was implemented for the survival analysis within the training set, yielding results that differ from those reported in the initial submission. In the revised analysis, features derived from multi-omic combinations no longer improved survival models. Instead, combining ex vivo drug response features (derived from CME) with clinical data enhanced model performance. When applied to the test dataset, these combined models also appear to outperform those based solely on clinical features. However, this conclusion can only be inferred from Figure 6B, as the corresponding numerical results are not reported in the text or supplementary tables. Including a concise description of model performance on the test data and discussing potential limitations would help place these findings into clearer context. Nevertheless, the revised analysis demonstrates improved rigor and statistical significance.

The description of the high-risk hyperdiploid subgroup that is associated with a high-methylation cluster and resistance to doxorubicin exposure has also been substantially improved and effectively integrated into Figure 4. With only minor additional clarifications, this manuscript appears suitable for publication.

Reviewer #2

(Remarks to the Author)

the authors responded to the comments accordingly, many thanks. Yet, the amendments in the abstract do not seem to be very helpful. What are the molecular dependencies the authors are talking about really? Still, some more concrete points would be helpful.

Reviewer #3

(Remarks to the Author)

The authors provided thorough and comprehensive responses to reviewer concerns. Their added emphasis on existing subtype classification is especially appreciated. Authors responded thoughtfully and made changes that improved clarity and emphasized clinically relevant conclusions. My concerns have all been adequately addressed.

Responses to Reviewers' comments: Olga Krali et al., manuscript #COMMSMED-25-0775

The line numbers outlined refer to the marked-up version.

Reviewer #1 (Remarks to the Author):

Krali et al. applied Multi-Omics Factor Analysis (MOFA) to integrate genomic, epigenomic, transcriptomic, and ex vivo drug response data from a large cohort of 1,231 pediatric patients with B-cell precursor acute lymphoblastic leukemia (BCP-ALL). This analysis identified ten key patterns, referred to as cross-modal elements (CMEs), which were subsequently used for pathway enrichment and intermodal network analyses. The impact of these CMEs on patient outcomes was assessed through survival modeling. Overall, the study offers valuable insights into multi-modal data integration and interpretation, with potential clinical relevance for patient stratification. The manuscript is well-structured, and the data are presented clearly and concisely.

We thank the reviewer for the summary of our work. Please see our point-by-point responses below.

Major comments:

A key concern is the moderate predictive performance of the proposed survival models. Baseline models using only clinical data achieved a concordance index (c-index) of 0.657 on the test set. The best-performing CMEf model reached a c-index of 0.699, which increased marginally to 0.703 when combined with clinical data (CMEf + Clinical). While these results suggest that integrating multi-modal data may improve outcome prediction, the conclusion would be more compelling if the authors demonstrated that the observed differences are significant. This could be addressed through stratified cross-validation, although it's important to recognize that CMEs may differ across iterations. Nevertheless, focusing on the best-performing model and the aggregated CME features from each iteration would allow for meaningful statistical testing.

We agree that this is an excellent suggestion. In response, we implemented repeated stratified cross-validation to evaluate the model performance using CME-derived features in relationship to the clinical baseline model. This resulted in 82 models (see results lines 399-429). The 82 models' cross-validated scores, standard deviation, 95% confidence interval, as well as Wilcoxon's signed-rank test p-values and BH adjusted p-values are provided in **Supplementary Data S14**. The result of this analysis pointed to that the ex vivo drug response measurements deriving from CMEs 2, 7 and 8 significantly improved the performance of the baseline clinical model.

These results are now reflected in the updated **Fig. 6**. Panel A shows the c-indices for the baseline clinical model and the three models where the information from CMEs 2, 7, or 8 significantly outperformed the clinical model (BH adjusted p-value < 0.05). Panel B

visualizes the c-indices in the test set for the clinical model and the three top performing models:

The methods (lines 203-206, 213-229), results (399-429), and discussion (lines 449-455) have been revised to reflect these changes.

Minor comments:

1) The title states “An integrative molecular map of pediatric B-cell leukemia” but reported is solely B-ALL.

We revised the title to “An integrative molecular map of pediatric B-cell precursor acute lymphoblastic leukemia”.

2) The Results section of the Abstract lacks specificity and does not clearly summarize the key findings presented later in the manuscript. Including more concrete outcomes would improve clarity and better reflect the study’s contributions. In Line 34: “This enabled the identification of molecular and drug dependencies relevant to patient risk stratification.” Please state one key example at least. Lines 35-37: most important survival model results should be stated with numbers.

The revised abstract now includes more concrete outcomes: “By leveraging molecular and drug dependencies, we stratified patients into subgroups that differ in relapse free survival. These signatures were independent of clinical variables. Survival models incorporating CME-selected ex-vivo drug responses combined with clinical data improved risk prediction compared to clinical models alone (FDR < 0.05), demonstrating the potential of integrative multiomics in refining risk stratification.” (lines 35-40).

3) Line 121: On what basis were the drug concentrations selected?

The drug concentrations were selected based on our previous work using FMCA (see Enblad et al., 2025, Hemasphere). The drug concentrations were selected empirically to maximise the variability of drug responses across pediatric BCP-ALL cells in this ex vivo culture system.

This is now clarified in the revised Methods Section lines 124-127 as following:

“The data represent the fraction of surviving leukemic cells after 72 hours of incubation with each drug at empirically selected concentrations (Supplementary Data S1). These concentrations were chosen to yield survival index (SI) values that capture inter-sample variability³⁸.”

4) Line 153: How were the top background CpG and genes defined? Based on their variance?

Yes, the background CpG sites and genes were selected based on the variance across all the samples in the training dataset. We clarified this in the Methods section (lines 141-143): **“To run MOFA, we selected a subset of CpG sites (n = 5,000) and genes (n = 10,000) with the highest variance in the training dataset, and included all of the available SNV (n = 529) and EVDR (n = 10) features.”**

5) Line 225: Table S2 contains a mixture of category labels (TRUE, 1, FALSE, 0), please unify.

We thank the reviewer for noticing this discrepancy. Table S2 (renamed to Supplementary Data S2) now only contains boolean values (True/False in Python and TRUE/FALSE in Excel).

6) Line 252: CME 2 had later the best performance for survival prediction and in addition to ETV6::RUNX1 also KMT2A and infants are significantly correlated. This is worth stating in the results.

After revising the survival modeling (see Major point, above), the CME2f (multiomics) model did not reach significance when evaluating if it outperformed the clinical based model. We have therefore omitted addressing this point in the revision.

7) Line 271: “and pathways with without overlapping” please remove “with”

The word **with** is removed in the new version.

8) Line 292: “Increased DNA methylation levels were associated with higher survival indexes (indicative of resistance) to both drugs (Dunn's test BH p-value <0.001, Fig. 4B-C). “What was the method applied to defining L and H patients? If clustering was used, please add a dendrogram to the heatmaps in Figure 4.

Unsupervised hierarchical clustering was used to define the two methylation groups shown in the heatmaps (Fig. 4B) and the violin plots (Fig. 4C). We revised the figure to show the previously omitted dendrograms.

We further clarified this in the Results section (lines 323–325): **“Unsupervised hierarchical clustering of DNAm from patients in the training set (n = 800), revealed two clusters of low and high DNAm levels (Fig. 4B).”**

9) Lines 299-311: The identification of a high-risk hyperdiploid subgroup is a particularly interesting finding and warrants greater emphasis. It would be valuable to include this

subgroup as an additional panel in Figure 4 to highlight its distinct characteristics. Additionally, please consider adding a subtype annotation column—at least for the most frequent subtypes—in the heatmaps presented in panels B and C. Furthermore, it would be helpful to clarify whether there are differences between the two hyperdiploid groups in terms of cohort distribution, patient age, clinical risk stratification, karyotypes, methylation patterns, or gene expression profiles. Lastly, are there any observed outcome differences between these groups in patients treated under more recent clinical protocols?

Based on this comment, we have now expanded the manuscript to include additional discussion about the HeH sub-groups (see Results lines 336-368 and Discussion lines 461-480).

The following changes were made to Fig. 4:

- Panel 4B (heatmaps) now contains the subtype annotation column (y-axis) to aid the interpretation of the observed patterns for both drugs (also **see panel B in point # 8, above**).
- In addition, subtype distribution plots from the previous version's Supplementary Fig. S5 were also added to main Fig 4D (also **see comment from Reviewer 3, point #3**):
- We added a Kaplan-Meier survival plot comparing the two HeH subgroups to **Fig. 4 (new panel 4F top)**, which was previously shown in Supplementary **Fig. S5**. We added results from Cox proportional hazards regression to adjust for potential confounders (treatment protocol and clinical risk group). This showed significant differential outcomes of the two HeH subgroups even after adjustment (Wald test p value = 0.04, HR = 2.34, 95% CI 1.04-5.2, **panel 4F bottom**).

To investigate whether there were co-variates associated with the two multi-omics network derived HeH clusters, we performed statistical and bioinformatics analyses looking at clinical and molecular variables available for these patients. In summary, no significant differences were observed between our two multi-omics network derived HeH clusters for:

- age distribution (Mann-Whitney U test p value = 0.59).
- white blood cell count (Mann-Whitney U test p value = 0.22).
- MRD response at day 29 for 78 HeH patients with MRD data available (Mann-Whitney U test p value = 0.36).

Furthermore, we annotated the chromosomal gains in each of the HeH patients in the training dataset (n = 240) using the raw intensities from 450k arrays using the R package *conumee* 2.0. By utilizing the HeH good/bad prognosis karyotype scheme outlined by Enshaei et al. (2021, *Lancet Hematology*), we annotated our HeH to good prognosis karyotype if +17 and +18, or +17 or +18 but lacking +5 or + 20. All remaining patients were annotated as poor prognosis karyotype. Again, in this analysis, no significant difference was observed between the two methylation groups in regards to good and bad prognosis karyotypes (fisher's exact test p-value = 0.22). The results of this analysis are described in the Results section lines 362-365.

RNA-seq data were available for 32 out of the 240 HeH patients in the training dataset: 8 patients in the methylation/doxorubicin-high group and 24 in the methylation/doxorubicin-low group. We performed differential expression analysis using the *limma* package and identified three differentially expressed genes. Specifically, *CLIP4* (log₂FC 1.14, FDR 0.035) and *ABCB1* (log₂FC 1.2, FDR 0.049) were overexpressed in the methylation/doxorubicin-high group, while *CYBB* (log₂FC -2.64, FDR 0.043) was downregulated. The results of this analysis are described in the Results section lines 365-368.

Given the strong literature around these genes, the findings suggest biologically distinct states in these two HeH subgroups. In particular, *ABCB1* encodes P-glycoprotein, a well-known efflux transporter implicated in multidrug resistance, including resistance to doxorubicin in multiple cancer types. These results reinforce our suspicion that the methylation/doxorubicin-high cluster corresponds to a drug-refractory phenotype. We added this to the discussion on lines 462-469:

“Differential gene expression analysis between the two HeH subgroups revealed upregulation of CLIP4 and ABCB1 and downregulation of CYBB in patients with a hypermethylation pattern and high doxorubicin SI%. ABCB1, in particular, which encodes the efflux pump P-glycoprotein, is a key drug-efflux transporter and well-known driver of anthracycline resistance⁶¹, suggesting a drug-refractory cellular state in this group. These results further support the presence of molecular subgroups within the HeH subtype that exhibit differential clinical outcomes, consistent with previous studies^{57–59,62}.”

10) Line 352: *“Combining CMEs with clinical data improved prognostic performance.” Is this significant?*

Please find our detailed response in the major comment #1 above.

11) Fig 4A. *What value determines the icon size?*

We have revised the figure caption to explain the icon size: ***“The size of each correlation (edge) is based on the correlation value, while the size of each circle (vertex) represents the number of connections + a weight of 10”*** (lines 758-759).

Reviewer #2 (Remarks to the Author):

In this manuscript, Kralli et al describe an integrative analysis of pediatric B-cell precursor ALL, based on retrospective DNA methylation, gene expression, SNVs and drug sensitivity data combined with multi-omic factor analysis (MOFA). This is a presentation of a great dataset for integrative analyses. Perhaps the most interesting finding is the identification of a high risk group of patients based on DNA methylation within the high hyperdiploid ALL group, commonly known as good outcome subgroup. This subcluster of patients with high DNA methylation patterns show poorer response to doxorubicin and etoposide.

We agree that the high hyperdiploid ALL subgroup is a very interesting finding, hence we have expanded the manuscript towards this direction and revised the respective sections (specifically, please see lines 35-39, 336-368 and Discussion lines 461-480).

The authors claim that their identification of so-called cross-modal elements and subsequent incorporation into survival models improved risk stratification. However, according to the data in the last figure 6, similar predictions in the test set were achieved between baseline models using clinical data only or such integrative CME analyses. Thus, it is unclear what the conclusion now really is, what should be done now in terms of molecular and functional analysis?

In response to this comment as well as the comments from Reviewer 1, we revised how we evaluated the survival models (please see Reviewer 1, comment 1 for details). Additionally, we revised the Discussion to further elaborate on the clinical utility of the models as follows (lines 449-455):

“Survival models incorporating CME-derived drug response features significantly improved predictive accuracy compared with traditional clinical risk stratification alone (FDR <0.05). Although the integrative CMEf and CMEf-all models did not outperform the clinical baseline overall, our framework proved effective for uncovering informative feature combinations. The ex-vivo drug-response signatures associated with specific CMEs offered added prognostic value, highlighting their potential in future risk stratification strategies”

Next to this, one major point here is, what is the congruence and/or relation to current risk assessment strategies, in particular CNA-based analyses (Moorman et al., blood 2014), or MRD-based risk stratification as use in current protocols?

All patients included in our study were treated according to the NOPHO ALL-92, NOPHO ALL-2000, or NOPHO ALL-2008 protocols. The NOPHO-92 and NOPHO-2000 protocols used similar genetic risk stratification schemes and did not include MRD-based risk assessment. MRD was implemented for the first time in NOPHO-2008.

Because MRD was part of the NOPHO-2008 protocol, we indirectly accounted for it by including risk groups as covariates in the baseline clinical model. This approach allows us to partially account for MRD information in the subset of patients where it was available.

Genome-wide chromosomal copy-number changes and large structural alterations can be inferred from DNA methylation (450k) data. However, a limitation using 450k arrays for CNA-based analyses primarily concerns small, focal alterations (e.g., *IKZF1* deletions), which require locus-specific assays like MLPA, SNP arrays, or WGS for accurate detection. We have previously explored focused-locus CNA inference from 450k arrays and observed variable concordance with MLPA and SNP arrays (unpublished observations), likely due to probe design where the 450k probes are enriched in CpG islands near gene promoters rather than evenly distributed across genes, and therefore are not optimized for reliable detection of focal copy-number changes.

We acknowledge that the inclusion of patients treated on earlier treatment protocols represents a limitation, as they do not fully reflect the current genetic and MRD-based risk stratification strategies. However, this historical cohort offers a major advantage: exceptionally long-term clinical follow-up, enabling robust outcome analyses that are rarely feasible in more recently collected cohorts. We explicitly noted this as a study limitation in the Discussion (lines 548–558).

Other major points:

- In Fig 2E-H, association of CMEs with clinical parameters and genetic subtypes – what are the p-values here?

The p-values in Fig. 2E (previously Fig. 2E-H) denote the Pearson's correlation coefficient (ρ) Benjamini-Hochberg (BH) adjusted p-values. The ρ and corresponding adjusted p-values are provided in Supplementary Data S3.

- In line 255, it is stated that CME 1, 2, 4 and 9 were associated with known clinical parameters,

a) which are these, what are the data?

In the revised manuscript on lines 281-283, we explicitly state the clinical parameters and data:

“While CMEs 1, 2, 4, and 9 showed significant associations with known clinical parameters, including molecular subtype, age, treatment protocol, and risk group, the remaining CMEs exhibited more subtle and complex interactions across modalities and clinical features, warranting further exploration.”

b) In Fig. 2E it is unclear what status and relapse describe. What does the size of the correspond to circles encode (p-value? legend scale?). Consider using a uniform representation.

The variable “status” denotes whether a patient was alive or deceased at the time of censoring (binary), and “relapse” indicates whether the patient experienced a relapse (binary). We agree that this was unclear in the previous version and have therefore explained the label status in the figure legend for clarity.

In the original figure, the size of each circle represented the absolute value of the Pearson correlation coefficient ($|\rho|$), such that larger circles corresponded to stronger correlations.

The color scale encoded the direction and magnitude of the correlation (dark blue = strong negative correlation; dark red = strong positive correlation).

We recognize that the *corrplot* visualization, where both color and size encode correlation strength, may have been confusing. To improve readability and ensure visual consistency across panels (Fig. 2E–H), we have replaced Fig. 2E with a heatmap representation in the revised version of the manuscript.

- Can the authors elaborate why SNVs and deletions in genes like *IKZF1*, *Pax5* and *TP53* do not contribute to the variance? Is *Pax5* alteration status (CME5/9) fully captured by other modalities?

SNV data were available for only a small subset of patients in the training (n = 96) and test (n = 32) datasets, which substantially limited the statistical power of this data modality. Moreover, the data were highly sparse, most of the 529 genes were mutated in only a few patients, resulting in a binary matrix dominated by zeros. This sparsity, combined with the low sample size, constrained MOFA's ability to detect meaningful patterns of unique or shared variance driven by SNVs.

As shown in Fig. 2E, with the exception of CMEs 5 and 9, none of the other CMEs capture variability in *PAX5*-altered cases. CME1 and CME2 separated patients with *ETV6::RUNX1*, high hyperdiploid, and *KMT2A*-rearranged subtypes into distinct clusters, whereas *PAX5alt* cases largely overlap with other subtypes. This indicates that *PAX5* alterations are only partially represented by other modalities in our model.

- The idea of a training and a test set would be to train one set and to analyze these results then in the test set – yet it is unclear throughout the manuscript what analysis was done in the training set and what in the test set, and what on the whole dataset.

We have clarified throughout the manuscript which analyses were performed on the training and test datasets. The Methods section has been revised accordingly.

Training dataset: used to train the MOFA model; perform enrichment, correlation, and network analyses; train survival models; and generate the main **Fig. 2–5, 6a** and **Supplementary Fig. S1a–S5**.

Test dataset: used exclusively for validation, including assessing the reproducibility of network dependencies (**Supplementary Data S12–S13**), generating an independent MOFA model for data imputation, evaluating survival model performance, and creating **Fig. 6b** and **Supplementary Fig. S1b**.

Fig. 1 provides an overview of the complete patient cohort and corresponds to the introductory results section (“Cohort demographics and data overview”). **Supplementary Fig. S1** presents the same information separately for the training and test sets.

Importantly, the training and test datasets were never combined in any analysis. This strict separation ensured that information from the test data did not bias model training or feature selection, preserving the integrity and generalizability of our results.

We have now added the following clarifying statements in the Methods: :

- **“MOFA was employed using the mofa2 R library²² on the training dataset” (line 140).**
- **“MOFA was used to impute missing values in the training set. A separate MOFA model was built for the test set using the same parameters as the training model, applied to impute missing values independently.” (lines 155-158).**
- **“Over-representation analysis (ORA) was performed on the top-ranked CpG sites and genes derived from the MOFA model trained on the training dataset” (lines 167-169).**
- **“For each CME, the most impactful features (absolute weight >0.6) were retrieved for both training and test datasets. Networks were generated from the training data, and reproducibility of central dependencies was assessed in the test data.” (lines 182-185).**

- It is unclear

a) what the survival indexes (Figure 4C) indicate, survival at what drug concentration, based on what parameter? What is the fraction of surviving cells relating to? Negative control (e.g. DMSO) at the endpoint (72h) or fraction of cells dying over time?

The survival index (SI) reflects the fraction of leukemic cells that remain viable after 72 hours of exposure to chemotherapeutic agents such as etoposide and doxorubicin (**Fig. 4C**). The specific concentrations used for each drug are listed in **Supplementary Data S1**. The concentrations were determined and empirically selected based on previous work (Enblad et al., 2025, HemaSphere) to maximise the variability in drug responses across samples, thereby enabling the detection of differences in response. A high SI indicates resistance, while a low denotes sensitivity.

We have clarified this in the Methods Section in lines **124-131**:

“The data represent the fraction of surviving leukemic cells after 72 hours of incubation with each drug at empirically selected concentrations (Supplementary Data S1). These concentrations were chosen to yield survival index (SI) values that capture inter-sample variability³⁸. The SI was calculated as the mean fluorescence signal from wells containing leukemic cells with intact plasma membranes after drug exposure, divided by the mean fluorescence signal from wells containing untreated leukemic cells (control), after subtracting the background signal from blank wells (medium only). This ratio was multiplied by 100 to obtain SI%⁴³.”

b) What does *ex vivo* resistance to cytotoxic anthracyclines refer to? Is this a complete non-response of the cells? It seems that the boxplots differ only marginally. Is it justified to talk about *ex vivo* resistance here? Consider showing individual drug responses in the boxplot.

In the Results section, we originally stated that “**Increased DNA methylation levels were associated with higher survival indexes (indicative of resistance)**”. To avoid overinterpretation, we have now removed the phrase “**indicative of resistance**” and clarified that these findings reflect relative cell viability rather than absolute *ex vivo* resistance. Higher levels SI% values denote higher cell viability following drug exposure, but the differences are relative between groups, and do not indicate complete non-response.

To further improve clarity, we have replaced the terms *resistant/resistance* with *higher cell viability* or SI% throughout this section.

In addition, we revised **Fig. 4C** to display individual data points within the violin plots, to provide visualization of the distribution of drug responses across samples for each drug.

- Could the authors speculate- based on their integrative analysis, on novel drug targets potentially in different subgroups?

While our study was primarily focused on data integration to identify relevant features of BCP-ALL that could help the prognostication and risk stratification of patients, our findings do suggest potential novel therapeutic avenues in distinct BCP-ALL subgroups. For example, low expression of histone gene families (H3, H2A, H2B) was associated with

higher % of surviving cells after vincristine exposure (**Supplementary Fig. S6**), consistent with the idea that quiescent or slowly cycling leukemic cells are less sensitive to microtubule-targeting agents. This highlights the therapeutic challenge of treating non-dividing leukemic cells. Potential strategies include combining vincristine with agents active in quiescent cells, such as BCL2 inhibitors (venetoclax). Additionally, we found that high expression of AP-1 complex members (*FOSL1*, *FOSB*, *JUNB*) was linked to higher % of surviving cells after doxorubicin exposure. Inhibition of AP-1 directly has been studied extensively, since AP-1 is often upregulated in various cancers. However, direct inhibition has shown some anti-cancer effects but has not yet been fully successful. Pharmacological modulation of AP-1-related pathways (e.g., MAPK signaling inhibitors) may represent an alternative strategy and could be relevant for patients with high expression of AP-1 complex members. These subgroup-linked vulnerabilities demonstrate how integrative, cross-modal analyses can point to novel therapeutic strategies beyond conventional risk stratification. We reflect upon this in the discussion (see lines **456-480 and 483-517**).

- CME3 is labelled as asparaginase and dexamethasone hub – two of the most relevant drugs of induction therapy. One would expect these to be highly predictive – yet this appears not the case, what is the explanation?

In our CMEf 3 survival model, we retrieved the feature importance scores for each signature. A feature importance score denotes the contribution of a signature to the overall model performance. The higher the value is the most influential a feature is. We observed that both dexamethasone (feature importance 0.002) and asparaginase (feature importance 0.002) were not within the top most contributing features for the survival model. This observation may be partly explained by the limited availability of EVDR data for asparaginase (<60% of the total cohort), which likely reduces the ability of the model to extract meaningful predictive information for this drug, despite the inclusion of dexamethasone (**Supplementary Data S1**).

- In Figure 4E and H, the authors talk about low, intermediate and high expression levels, what does this mean, what difference in expression are we talking about here?

The “low,” “intermediate,” and “high” expression levels correspond to three clusters identified by unsupervised hierarchical clustering of gene expression values. Specifically, we used the `cut_tree` function from the `scipy.cluster` module to define the three clusters based on the dendrogram branch splits.

In the revised version of **Fig. 4**, each heatmap is accompanied by the dendrograms to illustrate this clustering. The heatmaps previously shown in **Figure 4E** and **H** have been moved to **Supplementary Fig. S6**, which now contains all related panels for CMEs 8 and 10.

To better illustrate the expression differences amongst the three clusters, we have also added violin plots showing gene expression levels (y-axis) grouped by the expression cluster (x-axis), with significance levels indicated (Dunn's test BH adjusted p-values: *** < 0.001, ** < 0.01, * < 0.05, ns: non-significant).

Please see the **Supplementary Fig. S5** below:

- It is not fully clear how the networks Figures in Fig3B-D should be interpreted. It is difficult to judge the number of connectivities between nodes. Consider indicating a scale for the connectivity strength of the edges.

To improve interpretability, we have now color-coded the edges in Fig. 3B-D based on the overlap coefficient and added a corresponding color scale below the panels.

Supplementary Fig. S2 has also been revised accordingly. To maintain visual clarity, the thickness has been kept consistent across all networks. The figure caption has also been updated to reflect these changes.

Previously, the connectivities between nodes (pathways) were visualized using the Jaccard coefficient (*number of overlapping genes/number of total genes*). In the revised version, we instead used the overlap coefficient

(number of overlapping genes/number of total genes of the smallest path). This change ensures that smaller pathways that are subsets of larger ones can reach a maximum coefficient of 1, thereby providing a more intuitive representation of pathway overlap.

Please see the relevant section of revised Fig. 3 below:

- The conclusions as well as the title and the abstract of the manuscript are fairly generic. Please consider being more precise and concrete about the relevance for precision medicine. What would be actionable clinical conclusions?

We have revised the title and abstract (as also suggested by Reviewer 1) to better reflect the study's implications for precision medicine, highlighting how integrative multi-omics analysis can identify clinically meaningful patient subgroups and molecular features associated with therapy response.

In the Discussion (lines 461-480 and 497-498), we have expanded on the potential actionable insights, including how specific molecular patterns identified through multi-modal integration could support improved risk stratification and guide individualized treatment decisions in BCP-ALL.

Reviewer #3 (Remarks to the Author):

In this study, Kralli et al. combine SNV, gene expression, DNA methylation, and ex vivo drug response data through Multi-Omics Factor Analysis (MOFA) to characterize data from 1231 pediatric patients with BCP-ALL. This led to the identification of ten cross-modal elements (CMEs) explaining >2% of the variance in the training dataset. Each CME was thoroughly profiled, identifying key features and pathways. Functional modules for each CME were identified, within and across datasets, and gene expression and DNA methylation were

associated with particular drug responses in CME-specific networks. Finally, individual CMEs were incorporated into survival models, some of which slightly outperformed a clinical baseline model.

This work represents a thorough multi-omic characterization of pediatric BCP-ALL. Using the CMEs, authors have identified valuable connections between DNA methylation and gene expression involved in cancer-related processes, as well as informative interactions of both of those modalities with drug response data. These findings identify novel relationships that (if functionally validated) may lead to significant improvements in patient stratification and treatment selection.

One facet that was not thoroughly explored in this manuscript is the relationship of CMEs, key features of each CME, and molecular subtypes. Particularly since subtypes are used to stratify pediatric BCP-ALL patients and inform the use of targeted therapies, a subtype-focused exploration would be highly clinically relevant. Additionally, the incorporation of CMEs into survival models is of limited clinical utility. Since performing all (or in some cases any) of the assays used to identify CMEs is impractical at the bedside, the slight improvement in model performance with the addition of CMEs is an academic exercise. The survival modeling would be significantly more impactful if a handful of features (which could potentially be assayed clinically) were extracted from a CME of interest and their addition to the baseline clinical model improved performance.

Overall, the methodology in this work is sound and well-considered. Significant analysis has revealed novel interactions between gene expression, DNA methylation, and drug response. However, the lack of subtype-specific querying of the data and minimal clinical utility of the CME-based survival models limit the impact of this work on the characterization and treatment of patients with BCP-ALL.

We thank the reviewer for the comments and the observations. We agree that the subtypes are very important, hence we have expanded the manuscript towards this direction and revised the respective sections.

In addition, we addressed the issue that arises from the use of CME-based survival modelling to improve their clinical utility, by incorporating cross validation to perform statistical testing and retain only models that are significantly outperforming the baseline clinical model. The resulting three models included only a small number (<10) of top-ranked features (i.e. drugs), which can be potentially assayed in the clinics. If the selected models contained a large number of features, their number could be limited using the feature importance deriving from the gradient boosted survival models.

For a more thorough explanation, please see below our point-by-point responses.

Major points:

1. *Since the authors observe strong correlation of CME1 and 2 with molecular subtype, it would be beneficial to show a (PCA-style) plot of patient data across CME1 and CME2 (similar to the MOFA publication (Argelaguet et al., 2018)'s Figure 2E), with patients colored according to molecular subtype.*

In the revised version, we have added a scatterplot (updated **Fig. 2F**) displaying CME1 (x-axis) versus CME2 (y-axis), with patients color-coded by molecular subtype. To aid visualization, ellipses highlight molecular subtypes that show distinct CME1-CME2 distributions. The corresponding figure legend has been updated accordingly.

2. In the section "CMEs are enriched for cellular functions and regulatory networks," the statement that "Of these [genes in table S6], 36 genes were shared within the same CME". As written, this sentence suggests that 36 of the 1027 genes were enriched in one particular CME. Only by exploring the supplementary data does it become clear that the genes and CpG sites associated with those genes are top ranked features within the same CME, which alters the significance of the section. The transition within the same paragraph to CME-specific pathway enrichment deepens this confusion.

We have revised the text (lines 289-291) to accurately convey that the 36 genes and their corresponding CpG sites represent top-ranked features within the same CME, rather than genes enriched in a single CME:

"Of these, 64 top-ranked CpG sites were annotated to 36 top-ranked genes within the same CME, indicating a possible relationship between methylation level and transcriptional activity for these genes."

We also improved the transition to the subsequent enrichment analysis (lines 294-298), to ensure a smoother logical flow:

“To identify biological pathways enriched within each CME, we performed ORA using the CME-associated genes. Because the overlap between genes and annotated CpG sites was minimal, ORA was performed separately for each modality. This analysis revealed enrichment in 195 pathways and seven curated gene lists built based on prior knowledge^{37,46–49} (FDR < 0.01, Supplementary Data S8).”

3. In Figure S5A-B, the scales are unclear - does "percentage" represent the percentage of patients in the cluster of a particular subtype, or the percentage of patients with a particular subtype who fall into that cluster? A stacked bar plot could alleviate some of this confusion. This is especially significant because a large portion of the section "Inter-modal interactions within CMEs" is spent discussing low- vs. high-methylation groups of patients belonging to either ETV6::RUNX1 or HeH subtypes, but it's difficult to ascertain whether these observations are subtype-specific or not, and Figure S5 does not adequately clarify the question. Notably, both of these subtypes tend to have good outcomes, so further discussion of other subtypes would be appreciated. Clarity of this question in Figure 4 could be improved by adding a column to the left of the heatmaps (in panels B, E, and H) colored according to patient subtype, alongside the column colored according to L/H or L/I/H categorization.

Supplementary Figure S5A-B visualized the percentage of patients stratified by cluster who fall into each of the subtypes (e.g. the total percentage of the high cluster sums to 100%). We agree that this was confusing, and we updated the panels in the revised version to visualize the percentage of patients with a particular subtype who fall into the high or low methylation cluster (e.g. total percentage of HeH sums to 100%) using stacked bar plots to improve clarity. The subtype distribution plots are now part of main **Fig. 4** as panel **D** with additional subtype annotation next to the L/H categorizations in **Fig. 4B**.

Fig. 4D illustrates that while the etoposide-based clustering is mainly driven by the ETV6::RUNX1 subtype, the doxorubicin-based clustering spans subtypes, which is now more visible in the revised version of this plot.

This is now discussed in the Results section lines **331-341**: *“The low-methylation etoposide-associated cluster was predominantly composed of ETV6::RUNX1-positive patients (83.9%, n = 188), indicating that this subtype is closely associated with the methylation status of the CpG sites in the etoposide hub. In contrast, the doxorubicin hub spanned ALL subtypes, but with distinct subtype distribution. The low-methylation group was enriched for subtypes with generally favorable prognosis, including ETV6::RUNX1 (82.1%, n = 184), ETV6::RUNX1-like (66.7%, n = 10), high hyperdiploidy (HeH, 69.2%, n = 166), and low HeH (100%, n = 6). In comparison, subtypes associated with poorer outcomes, such as KMT2A-r (100%, n = 39), BCR::ABL1 (77.8%, n = 14), and BCR::ABL1-like (85.7%, n = 18), predominantly clustered within the high-methylation group.”*

D

Lastly, we added subtype and group annotations on the y-axis of the heatmaps in **Fig. 4B** and Supplementary **Fig. S5 b** and **f** (previously **Fig. 4E** and **H**).

Minor points:

1. Authors state in the MOFA section of Methods that "a subset of the most variable 5,000 CpG sites and 10,000 genes were used." How were these numbers of features selected? Considering the complexity of the disease and multiple subtypes, it's possible that a feature of interest may not have been included in the analysis. Were any features outside of this set evaluated or considered?

The subset of CpG sites (5,000) and genes (10,000) was selected based on the highest variance across all samples in the training dataset. We have clarified this in the revised Methods section (lines 141-147):

"To run MOFA, we selected a subset of CpG sites ($n = 5,000$) and genes ($n = 10,000$) with the highest variance in the training dataset, and included all of the available SNV ($n = 529$) and EVDR ($n = 10$) features. This variance-based feature selection follows MOFA recommendations to reduce dimensionality and exclude low-variance features. Because RNA-seq data were available for only 24% of patients compared to 87% for DNA methylation, we included a larger number of genes to balance the relative contribution of the transcriptomic modality and prevent overrepresentation of methylation data."

This approach follows recommendations from the MOFA documentation, which suggests removing low-variance features to reduce model complexity and facilitate convergence,

particularly when modalities differ greatly in dimensionality. Including only the most variable features ensures that the model focuses on biologically informative signals while avoiding domination of the factors by high-dimensional data such as DNA methylation.

In our training dataset, RNA-seq data were available for only 24% of patients, whereas DNA methylation data were available for 87%. To mitigate this imbalance and ensure sufficient representation of transcriptomic information, we included a larger number of genes (10,000) compared to CpG sites (5,000). This choice helped balance the contribution of each modality and preserve biologically informative variability in the transcriptomic data.

We acknowledge that this feature selection strategy may exclude genes relevant to rare or underrepresented subtypes. However, our main goal was to identify shared molecular patterns across the cohort rather than subtype-specific events. For this reason, no features outside this subset were included in the integrative MOFA analysis.

2. In the Network analysis section of Methods, how was the absolute weight >0.6 selected as the threshold for most impactful features? Do the biological insights fundamentally change with a cutoff of 0.5 or 0.7?

The absolute weight cut-off of > 0.6 was chosen empirically to balance interpretability and biological relevance, and was applied consistently across all downstream analyses, including the network analysis. A higher threshold (e.g., > 0.7) would substantially reduce the number of retained features, potentially omitting informative signals, whereas a lower threshold (e.g., > 0.5) would include more features at the cost of introducing noise. We applied the same threshold across all 10 CMEs to ensure compatibility of feature influence across the CMEs.

3. In figure 2E, which clinical variable does 'Status' represent? Also, what does the size of each dot represent?

As detailed in our response to Reviewer 2 (Major point b), “Status” indicates whether a patient was alive or deceased at censoring (binary variable). This is now clarified in the figure legend. The size of each dot in the original figure represented the absolute value of the Pearson correlation coefficient ($|\rho|$), where larger dots indicate stronger correlations.

Following both reviewers’ suggestions, we have updated **Fig. 2E** for improved clarity: the label “Status” and “Relapse” are clarified in the figure legend, and the panel has been replaced with a heatmap to provide a uniform and more interpretable visualization.

4. In addition to pathway enrichment separately across data modalities for each CME, authors may consider an integrated approach such as that offered by ActivePathways (Paczkowska et al., 2020).

We appreciate the reviewer’s suggestion to use ActivePathways for integrated pathway analysis. However, there is a fundamental methodological incompatibility between

ActivePathways and our MOFA-based approach. ActivePathways requires p-values from statistical group-wise comparisons (e.g., differential expression tests) as input, while MOFA produces unsupervised dimensionality reduction factor weights without predefined group comparisons. Thus, the two frameworks operate under distinct analytical paradigms: ActivePathways integrates significance evidence from hypothesis testing across datasets, whereas MOFA identifies latent factors explaining shared variance.

To nevertheless explore this possibility, we adapted our MOFA results for ActivePathways analysis by converting CME feature weights into pseudo p-values through permutation testing and ran ActivePathways using Gene Ontology and Hallmarks databases (with and without directionality) following Paczkowska et al., 2020, *Nature Communications* and Slobodyanyuk et al., 2024, *Nature Communications*. The analysis revealed minimal enrichment, likely due to data sparsity and limited overlap between the modalities (n = 36 genes). Only one significant pathway, ***TNF-alpha Signaling via NF-kB*** (CME 10, GEX modality), was identified, which was also recovered in our separate ORA analysis (**Supplementary Data S8**).

This outcome is consistent with prior robustness analyses of ActivePathways, which show sensitivity to sparse input and strict significance thresholds (Paczkowska et al., 2020).

Given that the ActivePathways approach did not yield any additional insight, we opted to omit this analysis from the manuscript.

5. In the first paragraph of the Discussion, authors claim that "Survival models incorporating CME-based features demonstrated a significant improvement in predictive accuracy compared to traditional clinical risk stratification." Given the c-index of best models (0.707) compared to the baseline clinical model (0.657), some statistical support for the claim of "significant improvement" would be beneficial.

As described in detail in our response to Reviewer 1 (Major comments), we revised the survival modeling framework to include repeated stratified cross-validation and statistical comparison of c-index values using the Wilcoxon signed-rank test with Benjamini–Hochberg correction. The result of this analysis (**Fig. 6** and **Supplementary Data S14**) pointed to that the *ex vivo* drug response measurements deriving from CMEs 2, 7, and 8 significantly improved the performance of the baseline clinical model. The Methods, Results, and Discussion sections have been updated accordingly (lines **213–229**, **397–429**, and **449–455**).

Additional changes by the authors are mentioned on the Cover letter

Responses to Reviewers' comments: Olga Krali et al., manuscript #COMMSMED-25-0775A

The line numbers outlined refer to the marked-up version.

Reviewer #1 (Remarks to the Author):

The authors have thoroughly revised the manuscript and successfully addressed all major concerns raised by the reviewers. The technical description of the ex vivo drug response profiling has been improved, enabling the reader to clearly understand what (and how) drug concentrations were selected and responses evaluated. For the MOFA analysis, the authors have now included a detailed description of the initial feature selection steps and have explicitly stated throughout the manuscript which analyses were performed on the training versus test data.

A cross-validation approach was implemented for the survival analysis within the training set, yielding results that differ from those reported in the initial submission. In the revised analysis, features derived from multi-omic combinations no longer improved survival models. Instead, combining ex vivo drug response features (derived from CME) with clinical data enhanced model performance. When applied to the test dataset, these combined models also appear to outperform those based solely on clinical features. However, this conclusion can only be inferred from Figure 6B, as the corresponding numerical results are not reported in the text or supplementary tables. Including a concise description of model performance on the test data and discussing potential limitations would help place these findings into clearer context. Nevertheless, the revised analysis demonstrates improved rigor and statistical significance.

The description of the high-risk hyperdiploid subgroup that is associated with a high-methylation cluster and resistance to doxorubicin exposure has also been substantially improved and effectively integrated into Figure 4. With only minor additional clarifications, this manuscript appears suitable for publication.

We would like to thank the reviewer for the constructive feedback.

In response, we added the numerical c-index test scores that are shown in Figure 6B to the Results section (lines 410-411 in the marked-up version) and to Supplementary Data S16 (Source Data Fig. 6A-B).

We further elaborated the limitations in the Discussion section (lines 426-430 marked-up version):

"Although our integrative CMEf and CMEf-all models did not generally outperform the clinical baseline, our framework proved effective for uncovering informative feature combinations. Our data demonstrated that survival models incorporating CME-derived drug response features (CMEs 2, 7, and 8) yielded added prognostic value. In this context, integrative multi-omics analyses primarily supported feature prioritization rather than directly enhancing the predictive performance of survival models."

Reviewer #2 (Remarks to the Author):

the authors responded to the comments accordingly, many thanks. Yet, the amendments in the abstract do not seem to be very helpful. What are the molecular dependencies the authors are talking about really? Still, some more concrete points would be helpful.

Thank you for the constructive feedback. In the revised version we provided additional clarification in the abstract about the “molecular dependencies” (Abstract result–Marked up version, lines 35-37).

"By leveraging ~~molecular and drug dependencies~~ correlations between DNA methylation and ex vivo response to doxorubicin, we stratify ~~ied~~ patients with hyperdiploidy into subgroups that differ in relapse free survival."

Reviewer #3 (Remarks to the Author):

The authors provided thorough and comprehensive responses to reviewer concerns. Their added emphasis on existing subtype classification is especially appreciated. Authors responded thoughtfully and made changes that improved clarity and emphasized clinically relevant conclusions. My concerns have all been adequately addressed.

Thank you for the valuable comments and suggestions throughout the revision process.